# Invariant Anomaly Detection under Distribution Shifts: A Causal Perspective

**João B. S. Carvalho, Mengtao Zhang, Robin Geyer, Carlos Cotrini, Joachim M. Buhmann**
Institute for Machine Learning
Department of Computer Science
ETH Zürich
{joao.carvalho, mengtao.zhang, robin.geyer, ccarlos, jbuhmann}@inf.ethz.ch

## Abstract

Anomaly detection (AD) is the machine learning task of identifying highly discrepant abnormal samples by solely relying on the consistency of the normal training samples. Under the constraints of a distribution shift, the assumption that training samples and test samples are drawn from the same distribution breaks down. In this work, by leveraging tools from causal inference we attempt to increase the resilience of anomaly detection models to different kinds of distribution shifts. We begin by elucidating a simple yet necessary statistical property that ensures invariant representations, which is critical for robust AD under both domain and covariate shifts. From this property, we derive a regularization term which, when minimized, leads to partial distribution invariance across environments. Through extensive experimental evaluation on both synthetic and real-world tasks, covering a range of six different AD methods, we demonstrated significant improvements in out-of-distribution performance. Under both covariate and domain shift, models regularized with our proposed term showed marked increased robustness. Code is available at: https://github.com/JoaoCarv/invariant-anomaly-detection.

## 1 Introduction

*Anomaly detectors* are the subject of increased interest in fields such as finance (Ahmed et al. [2016], Hilal et al. [2022]), medicine (Schlegl et al. [2019]), and security (Mothukuri et al. [2021], Siddiqui et al. [2019], Hosseinzadeh et al. [2021]). Having been trained on a sample from an unknown distribution, these models are capable of identifying abnormal objects unlikely to come from the original distribution (Bishop and Nasrabadi [2006]). Anomaly detection (AD) stands apart from supervised classification as it does not involve anomalies during training, making it challenging to articulate a model that depicts the class of objects deemed as normal.

AD as a field boasts a plethora of diverse methodologies (Ruff et al. [2021]). Current detectors have demonstrated the advantage of approaches based on representation learning (Reiss and Hoshen [2021], Deng and Li [2022]). In this context, an *encoder* maps objects to *representations* which capture the most distinctive features of an object. In addition, it strives to map the class of normal objects onto a subset characterized by a more regular shape, thereby rendering representations from abnormal samples easily identifiable by comparison. Central to this second goal is a notable vulnerability of representation learning-based methods: they hinge on the assumption of independent and identically distributed (i.i.d.) training and test data. This implies that normal samples in the training data are expected to be sampled identically in the test data, thereby being mapped to the same vicinity in the representation space - an assumption that is frequently violated in real-world scenarios (Koh et al. [2021]).

Indeed, distribution shifts in the context of AD present a unique challenge because it involves discerning two types of distribution shifts targeting the distribution of the normal objects, $p_n$. Anomalies

37th Conference on Neural Information Processing Systems (NeurIPS 2023).

can be conceptualized as samples from a different distribution $p_a$, which scarcely overlaps with the mass of $p_n$. This distribution $p_a$ can be interpreted as a shifted variant of $p_n$, and the task of anomaly detection becomes identifying this shift. Changes in the environment can also produce distribution shifts. When such a distribution shift transpires from the training data to the test data, $p_n$ is transformed into $\hat{p}_n$. If anomalies are naively perceived as merely samples from the original normal samples that are sufficiently different, then all samples from $\hat{p}_n$ could be wrongly classified as abnormal samples. Therefore, it is crucial to note that anomaly detectors need to identify shifted versions of $p_n$ that generate anomalies while being resilient to those originating from any other $\hat{p}_n$.

Additionally, anomaly detectors are particularly susceptible to changes in features that are incidentally correlated with being normal. This is because the training data consists only of normal objects, which in conjunction with consistent exposure to any recurrent features present in the environment, inherently facilitates the emergence of shortcut learning (Geirhos et al. [2020]). These effects can significantly impact the reliability of crucial anomaly detectors. For example, consider an anomaly detector designed for monitoring bank transactions, predominantly occurring during business hours. Such a detector could mistakenly hinge on the transaction time to determine anomalies. This approach could lead to fraudulent transactions executed at noon going undetected, while regular transactions made by individuals active at night might erroneously be flagged as suspicious. Thus, the effective detection of anomalies requires the model to go beyond such superficial correlations and instead focus on more substantive, causally relevant features.

In this work, we formalize the problem of anomaly detection in the presence of domain and covariate shifts using causality and information theory (Wang and Veitch [2022], Wang et al. [2022], Jiang and Veitch [2022], Liu et al. [2021], Linsker [1988], Stone [2004], Hjelm et al. [2018]). We view the generation of the dataset as a causal graph and distinguish between the environment that generates the dataset, the content features, the style features, and representations that are computed from these features. We note that to mitigate vulnerabilities to distribution shifts, we need to produce representations that are invariant to the environment where the object comes from (Veitch et al. [2021], Wang et al. [2022], Wang and Veitch [2022]). That is, the representation is not causally influenced by the environment. Through this formalization, we derive and formally justify a necessary condition for learning invariant representations within anomaly detection. To impose this condition, we introduce the partial conditional invariant regularization (PCIR) term for learning invariant representations in this setting. This term minimizes discrepancies between representations from different environments by minimizing discrepancies like the maximum mean discrepancy (MMD) (Gretton et al. [2012]). Similiar approaches relying on MMD regularization for domain generalization in different tasks have proven to be valuable (Kang et al. [2019], Long et al. [2013]). However, to our knowledge, all methods relying on conditional invariance have been applied under the assumption of access to all classes during training. This is not the case in the AD setting.

**Contributions:** **(1)** We conduct a series of experiments to illustrate the limitations of current AD methods in handling various distribution shifts. **(2)** Through causal inference we propose a novel formalization of the dual requirements of informativeness and invariance for robust AD models, leading to a new insight, as described in Theorem 2. **(3)** We introduce a regularization term to induce *partial conditional distribution invariance*, which serves as a strategic measure to ensure robust AD amid distribution shifts. **(4)** Our empirical findings demonstrate the effectiveness of our regularization term; all tested models exhibited an enhancement in performance under both covariate and domain shifts when augmented with *partial conditional invariant regularization* (PCIR).

## 2 Related Works

### 2.1 Distribution Invariant Representation Learning

The relationship between invariance and robustness to shifts in data distribution has been extensively explored, notably within the field of causal inference.

A range of studies has pursued the development of invariant representation learning, achieving notable success (Long et al. [2013], Kang et al. [2019], Mitrovic et al. [2020], Lv et al. [2022], Nguyen et al. [2021]). Moreover, a detailed examination of various forms of invariance-based methods derived from underlying causal graphs has been provided by Veitch et al. [2021] and Wang and Veitch [2022].

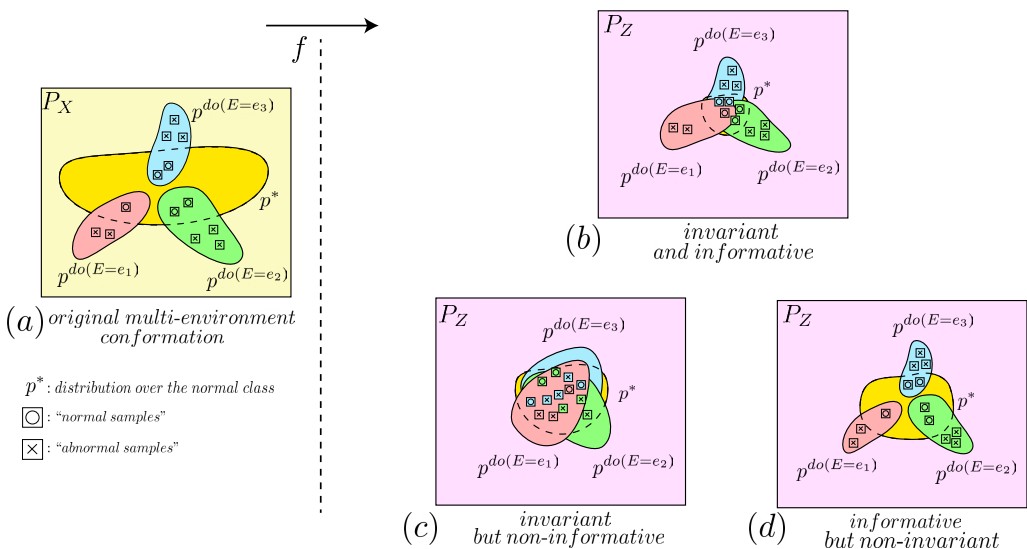

Figure 1: Comparative analysis of various mapping strategies from $\mathcal{X}$ to $\mathcal{Z}$ in the context of AD under distribution shifts. **(a)** Anomaly detection setting with multiple environments in $\mathcal{X}$. **(b)** Ideal scenario where the mapping is both invariant and informative; normal samples from different environments converge to the same subset of $\mathcal{Z}$, maintaining the distinction between normal and abnormal samples. **(c)** Mapping is invariant but not informative, resulting in both normal and abnormal representations collapsing to the same subset of $\mathcal{Z}$. **(d)** Mapping is informative but not invariant; although the mappings of all environments remain distinct, this makes the encoder vulnerable to distribution shifts.

Causal inference has also been shown to foster distributional robustness, as documented in Meinshausen [2018]. Building on this notion, *Invariant Risk Minimization* (IRM) (Arjovsky et al. [2019]) introduced a novel optimization principle hinged on maintaining invariance across diverse environments. However, Koh et al. [2021] demonstrated that this approach tends to falter when subjected to real-world distribution shifts. Yao et al. [2022] has introduced selective data augmentations to induce distribution shift robustness. However, due to its reliance on class labels, it is not directly applicable to AD.

## 2.2 Deep Anomaly Detection

A common strategy in AD involves the deployment of autoencoders. These methods operate by learning normal patterns within a dataset and then attempting to *reconstruct* new samples. The premise is that anomalous samples, due to their distribution differences, will yield higher reconstruction errors. Several studies have adopted this method, demonstrating its efficacy in a range of contexts (Gong et al. [2019],Park et al. [2020],Yan et al. [2021]).

In the wake of recent advancements in self-supervised representation learning (He et al. [2019], Chen et al. [2020]), *contrastive learning* has emerged as a viable method for AD. Approaches such as *PANDA* (Reiss et al. [2021]), and the more recent *MeanShift* (Reiss and Hoshen [2021]) and *CFA* (Lee et al. [2022]) harness these strategies to finetune pre-trained encoders to the AD setting. Conversely, *CSI* (Tack et al. [2020]) and *DROC* (Goyal et al. [2020]) use contrastive learning to learn a representation of the normal data from scratch. *Knowledge distillation*-based methods also hold a significant place in AD research. A student-teacher framework, which incorporates discriminative latent embeddings, was first introduced in Bergmann et al. [2020]. The most recent iterations, *Reverse Distillation* (Deng and Li [2022]) and *STFPM* (Wang et al. [2021]) have attained state-of-the-art performance across numerous AD tasks. Recently, *Red Panda* (Cohen et al. [2023]), has adapted *PANDA* by adding an additional disentanglement term to the loss derived from Kahana and Hoshen [2022], with the intent of inducing robustness to previously identified attributes.

Despite the effectiveness of these methods, it is important to note that they primarily focus on AD within datasets where the training and test data distributions are identical. As we will show, its performance tends to degrade when confronted with data that exhibit different kinds of distribution shifts.

One way to improve robustness to distribution shifts entails pretraining the encoder using an invariance-inducing method (for example, IRM or LISA), applied to an unrelated pretext task using the same data (Smeu et al. [2022]). Unfortunately, this requires the existence of additional task labels, which is uncommon for AD tasks. Moreover, as we experimentally demonstrate, this approach achieves limited performance when paired with state-of-the-art AD methods.

## 3 Formalization

### 3.1 Background

Our formulation operates under the assumption of a given set of *environments*, denoted as $\mathcal{E}$, and a sample space, $\mathcal{X}$. We have a sample $D = \{x_1, \ldots, x_n\}$ drawn from an unknown distribution $p_n$, whose mass is mostly concentrated on a subset $N \subseteq \mathcal{X}$. This subset, $N$, defines the *normal class*. Given that $X \sim p_n$, we consider $X$ as a pair of random variables $X_a$ and $X_e$, such that $X = (X_a, X_e)$. Here, the environment $E$ only influences $X_e$. Conceptually, $X_a$ represents the component of $X$ that determines whether $X$ is an anomaly, while $X_e$ comprises style features. These style features, while unaffected by the anomaly status of an object, are influenced by the environment. Note that this assumption implies that $\mathcal{X} = \mathcal{X}_a \times \mathcal{X}_e$, for some appropriate sets $\mathcal{X}_a$ and $\mathcal{X}_e$. Fig. 1 (a) illustrates our setting under three different environments.

Our main goal in this section is to elicit the requirements that representations should fulfil in order to lead to effective anomaly detectors. More precisely, we argue that representations must simultaneously (i) maximize the information they contain about the original objects and (ii) they must be invariant to the environment.

### 3.2 Requirements for Robust Encoders

We illustrate the necessity of invariance and informativeness for effective AD using an example with random variables $X_1, X_2, X_3$. We assume that they are distributed over the set $N$ of all points $(x_1, x_2, x_3) \in \mathbb{R}^3$ such that $0 \leq x_1 = x_2 \leq 1$. We assume that $x_3$ denotes the environment. Our dataset $D$ consists of points in $N$ where $x_3 = 0$.

Consider now the following encoders $f_1$, $f_2$, and $f_3$, where $f_1(x_1, x_2, x_3) = x_1$, $f_1(x_1, x_2, x_3) = (x_1, x_2)$, and $f_1(x_1, x_2, x_3) = (x_1, x_2, x_3)$. Note that encoder $f_1$ is invariant but lacks the necessary information to detect anomalies, as it can map different points to the same representation. Indeed, the normal point $(1, 1, 1)$ and the anomaly point $(1, 2, 1)$ would be mapped to the same representation, the number 1. Hence, any anomaly detector based on this encoder can be easily evaded. Encoder $f_3$ captures all information but is not invariant, which leads to a potential evasion of AD. Encoder $f_2$, on the other hand, is both invariant and informative, correctly capturing the necessary information to detect anomalies while ignoring the environment variable $x_3$.

This simplified scenario emphasizes the dual necessity of invariance and informativeness for the development of robust anomaly detectors. Regrettably, as we will demonstrate, contemporary AD methods tend to excel in informativeness while significantly lagging in invariance. For a graphical illustration of this effect, please see Fig. 1 (b-d).

We now formalize the desiderata for representations computed by an encoder for AD.

### 3.3 Invariant Representations

Let $f$ be an encoder mapping a $\mathcal{X}$ to a representation space $\mathcal{Z}$. We also let $Z = f(X)$ denote the *representation* of $X$.

**Definition 1.** We say that $Z$ is an *invariant representation* of $X$ under *domain shifts* if

$$p^{do(E=e)}(Z = \cdot) = p^{do(E=e')}(Z = \cdot), \quad \text{for any } e, e' \in \mathcal{E}. \tag{1}$$

**Definition 2.** We say that $Z$ is an *invariant representation* of $X$ under *covariate shifts* if

$$p^{do(X_e=x)}(Z=\cdot) = p^{do(X_e=x')}(Z=\cdot), \quad \text{for any } x, x' \in \mathcal{X}_e. \tag{2}$$

We simply say that $Z$ is an invariant representation of $X$ if it remains unaltered under both domain shifts and covariate shifts. Note that domain shifts are stronger than covariate shifts. Specifically, domain shifts entail assessing data from diverse sources with disparate parameters, whereas covariate shifts pertain to subtle alterations in the existing data. Consequently, the broader changes inherent to domain shifts usually exert a more substantial impact on data representation stability and are harder to tackle.

Invariant representations prevent anomaly detectors from incorrectly classifying objects that result from covariate and domain shifts. These representations are, by definition, resistant to changes in $E$ or $X_e$. As a result, adversaries cannot expect to alter the detection of an anomaly by merely using a different environment or changing style features of the anomaly.

However, solely enforcing invariant representations can be insufficient, and potentially lead to a collapse of the representations (see Fig. 1 (c)). For this reason, we argue that representations shall store as much information about the original object as possible. This principle is in line with the design of many popular encoders (Linsker [1988], Stone [2004], Hjelm et al. [2018]) and can be formalized using information theory.

### 3.4 Mutual Information between Representations and Objects

We can use the mutual information between two random variables as our metric to evaluate how much information about $X$ is captured by the representation $Z$.

**Definition 3.** The *mutual information* $I(X; Z)$ between $X$ and $Z$ is defined as

$$I(X; Z) = \int \int \frac{p_{X,Z}(x,z)}{p_X(x)p_Z(z)} dx\, dz. \tag{3}$$

Here, $p_{X,Z}$ is the joint pdf of $(X, Z)$ and $p_X$ and $p_Z$ are the marginal pdfs of $X$ and $Z$, respectively.

Intuitively, $I(X; Z)$ indicates how much information learning about one of $X$ or $Z$ reveals about the other. We argue then that encoders shall compute representations that keep as much information as possible from the original objects.

## 4 Learning Invariant and Informative Representations

Having outlined the prerequisites for anomaly detection representations, the next task is to devise strategies that enable encoders to learn both invariant and informative representations. Encouragingly, most contemporary AD methods inherently employ strategies that maximize the mutual information between the observation and the representation (Tishby and Zaslavsky [2015]). This is often achieved either via a reconstruction loss term (Kong et al. [2019]) or a contrastive loss term (Sordoni et al. [2021], Wu et al. [2020], Oord et al. [2018]).

We dedicate the rest of this section to developing a strategy suitable for invariant representation learning under the scope AD.

### 4.1 Learning Invariant Representations

We start by introducing a causal graph for anomaly detection in Fig. 2. In these graphs, $E$, $X_a$, and $X_s$ denote the environment, the relevant features, and the style features. We introduce two new random variables $W$ and $U$. $W$ is a binary variable indicating if the object is normal ($W = 0$) or an anomaly ($W = 1$). $U$ denotes all confounding factors. That is, factors that produce correlations between $X_a$ and $X_e$ during the sample generation process.

We now recall that an encoder $f$ that attains invariant representations is $X_a$-measurable (Veitch et al. [2021]). The following result, which follows from a theorem from Veitch et al. [2021], argues that we can ensure invariant representations by enforcing a possibly conditional statistical independence between $Z$ and $E$.

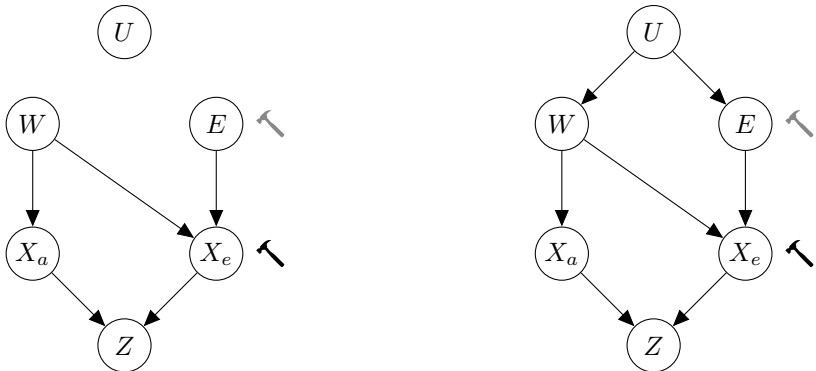

Figure 2: Causal graphs for anomaly detection. The left figure shows the case of no confounding. The right figure shows the case of confounding. An intervention at the $E$ variable induces a domain shift (gray hammer), whereas an intervention at the $X_e$ variable induces a covariate shift (black hammer).

**Theorem 1.** Suppose that $f$ learns invariant representations. If $W$ and $E$ are confounded, then $Z \perp E \mid W$. Otherwise, $Z \perp E$.

*Proof.* We sketch a proof here and provide a more rigorous proof in the appendix. Since $f$ learns invariant representations and is, therefore, measurable, we can say that $Z$ is $X_a$-measurable. This means that any probability of an event involving $Z$ can be rewritten as a probability of an event involving $X_a$. As a result, it suffices to show that $X_a \perp E \mid W$, when $W$ and $E$ are confounded and $X_a \perp E$, otherwise. These claims can be shown using d-separation. For example, when $W$ and $E$ are not confounded, there are only two paths from $X_a$ to $E$ and they are blocked. Hence, $X_a \perp E$ by d-separation. $\qquad\square$

The fundamental question we must address is how to ensure statistical independence between $Z$ and $E$. If such independence holds, we find that for all $e, e' \in \mathcal{E}$, it implies that $p^{do(E=e)}(Z = \cdot) = p^{do(E=e')}(Z = \cdot)$. In general, we cannot access counterfactual examples, so enforcing counterfactual invariance becomes impossible. However, it is still possible to induce a counterfactual invariance signature by imposing appropriate conditional independence conditions. In practice, this condition can be set as MMD $(p(Z = \cdot \mid E = e), p(Z = \cdot \mid E = e')) = 0$, where MMD stands for maximum mean discrepancy (Gretton et al. [2006, 2012]), and measures the distance between two distributions using empirical samples by calculating the divergence between the means of these sample sets once they have been projected into a reproducing kernel Hilbert space (RKHS).

### 4.2 Partial Conditional Invariant Regularization (PCIR)

Based on the previous insights, we propose using the MMD between two distributions as the driver for the invariance-inducing regularization term. Taking into account our previous formulation depicted in Fig. 2, we are now in a position to derive a novel regularization term specifically designed for invariant AD:

$$\Omega_{\text{PCIR}} = \sum_{\substack{e,e' \in \mathcal{E} \\ e \neq e'}} \text{MMD}\left(p(Z = \cdot \mid W = 0, E = e), p(Z = \cdot \mid W = 0, E = e')\right). \tag{4}$$

We call this approach *partial conditional invariant regularization* (PCIR), as it induces conditional distribution invariance over only one instantiation of $W$.

This partial conditional regularization aligns with other regularization terms proposed in preceding research (Veitch et al. [2021], Li et al. [2018]). It is 'conditional' because, in the event of confounding, we must condition on $W$ when computing the MMD. It is 'partial' due to the fact that the training dataset only contains samples where $W = 0$.

# 5 Experiments

## 5.1 Experimental Setup

Our solutions are validated under two distinct settings: domain generalization across domain shifts and domain generalization across covariate shifts. All experiments described were the result of 5 repetitions over different seeds. For additional details on the experimental design please refer to the supplementary material.

**Real-world domain shift**   For a realistic anomaly detection scenario, we considered the task of identifying tumorous tissue from images of histological cuts, using the Camelyon17 (Koh et al. [2021], Bandi et al. [2018]) dataset. This dataset consists of five different subsets of images arising from five different hospitals, with domain shifts occurring due to differences in slide staining, image acquisition protocol, and patient cohorts. This presents a challenging anomaly detection task, as the alterations that differentiate normal from abnormal samples are often subtle and may correlate with unknown features that are domain-dependent. In our experimental design, we motivate an environment per domain and set up two sets of experiments: one using three environments for training and another using two environments for training.

**Real-world shortcut**   We have also evaluated our method in the Waterbird dataset (Sagawa et al. [2019]). This is a real-world natural image dataset where the distribution shift occurs as the natural habitat depicted in the background changes between an aquatic and land setting. From the two kinds of birds in the dataset (water and land birds), water birds were assigned to training data and land birds were set as an anomaly. To make this a more challenging setup, we have defined a highly unbalanced distribution of the environments, with 184 images in training data with a water background and 3498 with a land background. The evaluation of the methods was performed in images with only a water background for out-of-distribution and only land background for in-distribution.

**Synthetic covariate shifts**   To evaluate the robustness against covariate shifts, we use the DiaViB-6 dataset Eulig et al. [2021] for our experiments. This dataset comprises modified and upsampled images from MNIST Deng [2012], where the generative factors of the image can be altered, leading to changes in texture, hue, lightness, position, and scale.

The training data is comprised of two distinct environments, generated by manipulating original instantiations of each factor, and ensuring all factors differed between the two environments. For the test data, we defined five distinct environments denoted as $e_0, e_1, ..., e_4$. The environment $e_0$ corresponds to images from MNIST for a specific digit under an original instantiation of each factor. All images of that digit are labeled as normal, while images of any other digit are considered anomalies. For $0 < i \leq 4$, each environment $e_i$ corresponds to images where $i$ factors have been modified. In $e_i$, all factors modified in $e_{i-1}$, plus an additional unique factor, are altered. The task is to classify an image of a handwritten digit as either normal or an anomaly.

We also adapted this setup to include the Fashion-MNIST dataset, a more challenging synthetic benchmark. The same cumulative covariate shifts were induced from an original in-distribution environment, $e_0$, to subsequent environments $e_{1-4}$. Two specific classes were chosen to generate normal and abnormal samples.

**Anomaly detection methods tested**   We undertook a comprehensive evaluation of our proposed regularization term, PCIR, and integrated it into six diverse methodologies for deep AD. These include: *STFM* Wang et al. [2021], *Reverse Distillation* Deng and Li [2022], *CFA* Lee et al. [2022], *MeanShifted* Reiss and Hoshen [2021], *CSI* Tack et al. [2020], and Red PANDA Cohen et al. [2023]. For a detailed exposition of these methods, refer to the supplementary material.

## 5.2 Results

**Real-world anomaly detection under domain shift**   Notably, even with the complexity and real-world variability introduced by the domain shift, the incorporation of partial conditional distribution invariance still resulted in notable improvements for both training setups, in particular with *MeanShift* and *Red PANDA* retrieving almost in-distribution performance (see Fig. 3). This suggests that our regularization term is robust to more substantial changes associated with domain shifts.

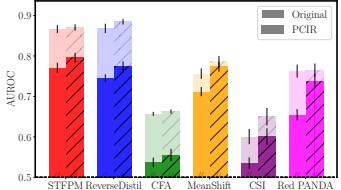 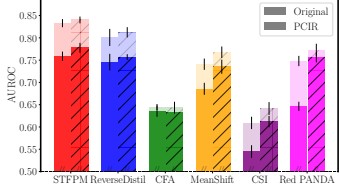 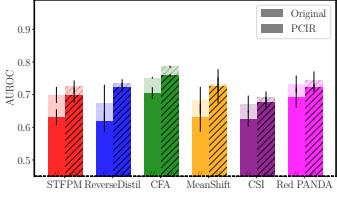

(a) Camelyon17: 2 environments     (b) Camelyon17: 3 environments     (c) Waterbirds

Figure 3: Results on realistic domain shift Camelyon17 and realist shortcut learning in Waterbirds. (**background transparent bar-plots:** in-distribution evaluation; **foreground opaque bar-plots:** out-of-distribution evaluation). **(a)** Camelyon17: Setup with two in-distribution training environments and three out-of-distribution test environments. **(b)** Camelyon17: Setup with three in-distribution training environments and two out-of-distribution test environments **(c)** Waterbirds: Shortcut learning setup.

Focusing on the out-of-distribution setting, our results highlight a consistent pattern of performance enhancement when applying the partial conditional invariance regularization term, ranging from 2% to 8% increase in the AUROC when comparing regularized and un-regularized methods. The consistency of this performance boost across different models and with both two and three in-distribution training environemnts underscores the potential of PCIR as a beneficial regularization technique in out-of-distribution settings.

**Realist shortcut learning** We observe that in all original unregularized methods evaluated, there is a noticeable drop in performance when comparing in-distribution (background transparent plots) to out-of-distribution (foreground opaque plots) that is attributed to the models effectively capturing the exposed shortcut. Models with PCIR nearly recover the in-distribution performance, showing the effectiveness of this approach to ignore uninformative environment features even when exposed through a shortcut. Furthermore, in all models tested, there's a consistent observation that the model performance not only stabilizes but also increases when adding the regularization term even for in-distribution scenarios. The increase in model performance, attributed to the addition of PCIR to each method, exhibits an improvement range from 5% to 15% AUROC.

**MMNIST and Fashion-MNIST under covariate shift** Models that absorb shortcut features have been observed to be especially susceptible to covariate shifts (Eulig et al. [2021], Geirhos et al. [2020]). Therefore, if a model abuses a shortcut, then inducing a covariate shift in the shortcut feature often significantly deteriorates performance. This effect is discernible in Fig. 4, where performance drops progressively increase for all models as more covariate shifts are induced in the images, and thus more shortcut features deviate from their original form in the training data.

However, note that by adopting regularization based on partial conditional invariance, we have been able to construct anomaly detectors that consistently exhibit enhanced robustness against induced covariate shifts. In some cases, even in-distribution performance increases through partial conditional invariance. These findings corroborate our hypothesis that partial conditional distribution invariance serves as a sufficient prior for robustness to covariate shifts in AD. For a more comprehensive analysis of performance improvements achieved refer to the supplementary material.

**Ablation study** In our ablation study, we held all method parameters constant to exclusively examine the impact of the weight assigned to the partial conditional invariance regularization term.

Across all tested methods, the relationship between regularization weight and performance exhibited a concave shape. This pattern suggests a simple linear search could be sufficient to identify an optimal weight for the regularization term. This observed behaviour aligns intuitively with the nature of invariance regularization. Over-emphasizing the regularization term could potentially cause the model's encoder to generate non-discriminative and non-informative representations. In an extreme case, the model could collapse all inputs into a single value. Interestingly, the behaviour of the regularization term remained consistent across both datasets used in our study. This reinforces the robustness of our approach across diverse datasets.

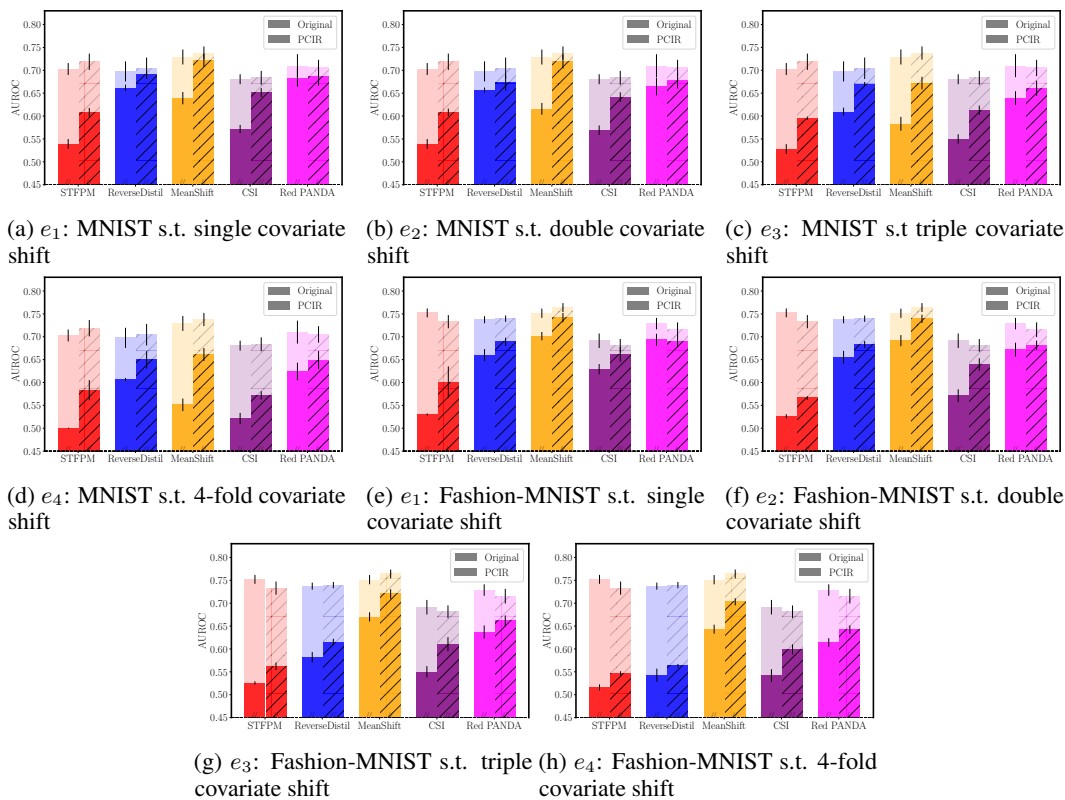

(a) $e_1$: MNIST s.t. single covariate shift

(b) $e_2$: MNIST s.t. double covariate shift

(c) $e_3$: MNIST s.t triple covariate shift

(d) $e_4$: MNIST s.t. 4-fold covariate shift

(e) $e_1$: Fashion-MNIST s.t. single covariate shift

(f) $e_2$: Fashion-MNIST s.t. double covariate shift

(g) $e_3$: Fashion-MNIST s.t. triple covariate shift

(h) $e_4$: Fashion-MNIST s.t. 4-fold covariate shift

Figure 4: Experimental results on synthetic covariate shift (**background transparent bar-plots:** in-distribution evaluation; **foreground opaque bar-plots:** out-of-distribution evaluation) **(a-d)** Results in MNIST. **(e-h)** Results in Fashion-MNIST.

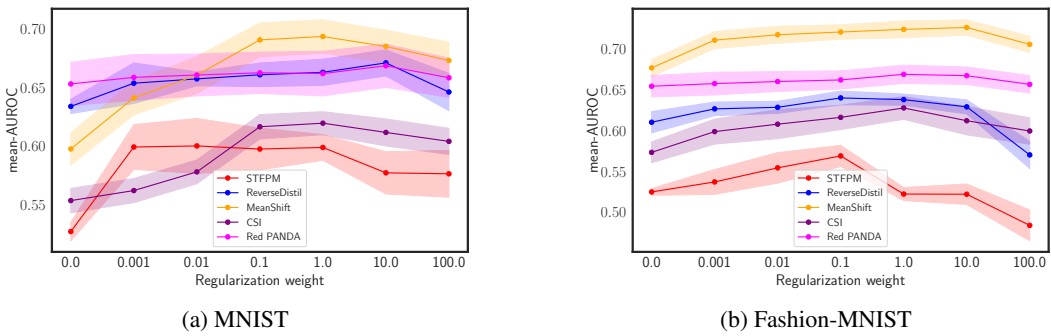

(a) MNIST

(b) Fashion-MNIST

Figure 5: Ablation study over the weight of the regularization term for MNIST and Fashion-MNIST under distribution shift. Regularization weight determines the weight of PCIR in the training loss.

## 6  Discussion and Limitation

**Invariance regularization also improves in-distribution AD**   A surprising result from this study was the consistent enhancement in in-distribution performance across most methods when partial conditional invariance was incorporated. This observation can be explained by recent advances suggesting that invariance also bolsters in-distribution robustness against noise and sampling variability (Lopes et al. [2019], Mitrovic et al. [2020]) - factors to which anomaly detection is inherently vulnerable (Ramotsoela et al. [2018]). By design, invariance regularizers discourage representations encoding style features. Hence, representations only carry over information that is inherently related to the object rather than the environment. This leads to models identifying meaningful patterns instead of noisy variations in the data, even in-distribution.

**Unlabeled environments** A prominent limitation of invariance regularization lies in its dependence on environment labels (Veitch et al. [2021], Jiang and Veitch [2022], Arjovsky et al. [2019]). Additionally, as evidenced in our experimental comparisons between covariate and domain shifts, interventions directly applied to the covariates during environment generation yield enhanced robustness under the constraints of partial conditional invariance. This underscores that conditional invariance regularization's effectiveness diminishes when the environments are not distinctly segregated. However, in situations where datasets lack explicit environment partitions, alternative strategies can be employed. As demonstrated by Lin et al. [2022] it remains feasible to jointly estimate environment partitions and invariant representations, provided there is access to sufficient auxiliary information. This finding opens the door to expanding the applicability of partial conditional invariance in AD, even in scenarios with limited information on environmental conditions or corrupted separation between environments.

**Invariance beyond partial conditioning** Theorem 1 shows that in the presence of confounding, learning invariant representations requires that representations are independent from the environment, when conditioned on $W$. Note that our partial conditional invariant regularization conditions only on $W = 0$ and not on $W = 1$ (see Equation 4). This is an inherent limitation due to the fact that we only have normal objects in the sample. However, our regularization is still powerful enough to provide improvements over the state of the art. We argue that the quality of this improvement depends on how disentangled content features are from style features (i.e., $X_a$ and $X_e$ in Figure 2). Consider, for example, an MNIST setting where normal objects are images of particular digit and images of any other digit are anomalies. Furthermore, assume that images from one environment have one background color and images from another environment have another background color. There is a clear disentanglement between style (i.e., the digit) and content (i.e., the background color). In such settings, attaining invariance to the background color just with $W = 0$ leads also to invariance to the background color with $W = 1$. We conjecture that partial conditional invariant regularization is sufficient when $W$ does not influence $X_e$; that is, when there is no arrow between these variables in Figure 2.

**Multi-shift environments** Certain datasets, such as the one described by Christie et al. [2018], take into account data shifts across two different kinds of domain shifts (e.g. time and location), equivalent to dual interventions in a bivariate $E$ in our formulation. While the extension of our work to address multi-attribute settings might be plausible in a dual intervention scenario, it presents a non-trivial challenge as the number of different possible interventions increases, given that it requires the induction of pairwise invariance across environments. This issue continues to be an area of active research.

**Fairness** One potential extension of this work concerns the setting where either a covariate shift is induced over a protected attribute (e.g., gender or race), or a domain shift that permeates to such attributes. Such nuances have been brought in the context of invariant representation learning by Veitch et al. [2021] and could serve as a potential application of this work in AD. In particular, we could see our regularization term be implemented similarly to Louizos et al. [2015], but instead with the intent of effectively discarding incidental correlations that might reflect societal or systemic biases present in the data.

## 7  Conclusions

Our study sheds light on the significant challenges anomaly detectors face in the context of distribution shifts. We proposed a novel, robust solution centered around invariant representations, which mitigates the impact of shortcut learning by enforcing statistical independence between the representations and the environment. Empirical validation of our theoretical proposals confirmed the effectiveness of our approach, with a regularization term inducing partial conditional distribution invariance, significantly improving model performance under covariate and domain shifts. We believe these findings pave the way for a deeper understanding of AD methods' robustness and how to mitigate their vulnerability to distribution shifts.

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

# A Theoretical Details

## A.1 Measurability

Let $\mathcal{B}(\mathbb{R}^n)$ be the Borel $\sigma$-algebra over $\mathbb{R}^n$, for some $n \in \mathbb{N}$.

**Definition 4.** The $\sigma$-algebra generated by a random variable $Y$ is $\sigma(Y) = \left\{Y^{-1}(B) : B \in \mathcal{B}(\mathbb{R}^n)\right\}$.

**Definition 5.** A random variable $X$ is $Y$-measurable if $\sigma(X) \subseteq \sigma(Y)$.

Intuitively, the $\sigma$-algebra generated by $Y$ describes all events that $Y$ "can express" and can be measured in probability. If $X$ is $Y$-measurable, that means that $Y$ can express all what $X$ can express.

## A.2 D-separation and Confounding

We provide here a brief overview on d-separation and confounding and refer the reader to Bishop and Nasrabadi [2006] for details.

**Definition 6.** Two random variables in a Bayesian network are confounded if they share a latent parent.

**Definition 7.** A *path* is a sequence of random variables $(V_1, \ldots, V_n)$ in a Bayesian network, where $V_i$ is a parent or child of $V_{i+1}$, for $i < n$. For $1 < i < n$, the variable $V_i$ is a *collision* if $V_i$ is a child of both $V_{i-1}$ and $V_{i+1}$.

**Definition 8.** A path is *blocked* if $V_1$ and $V_n$ are not observed and at least one of following holds:

- For some collision $V$ in the path, neither $V$ nor any of its descendants is observed.

- For some variable $V$ in the path that is not a collision, $V$ is observed.

We now recall the d-separation principle.

**Lemma 1.** Let $W$ be a set of observed variables and $V_1$ and $V_2$ two variables not observed. If any path from $V_1$ to $V_2$ is blocked, then $V_1 \perp V_2 \mid W$.

## A.3 Detailed Proof of Theorem 1

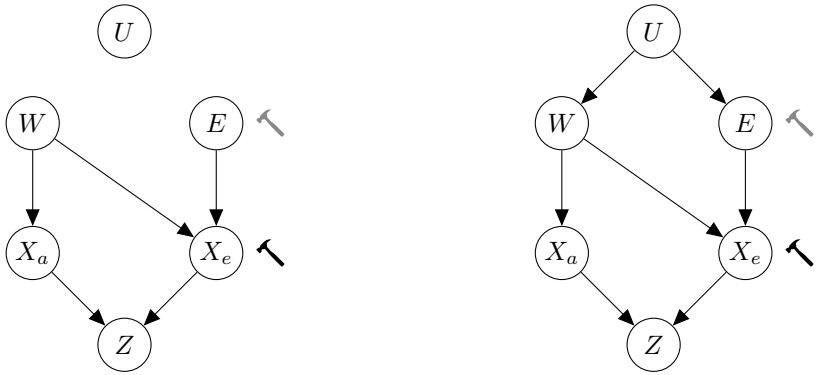

Figure 6: Causal graphs for anomaly detection. The left figure shows the case of no confounding. The right figure shows the case of confounding. An intervention at the $E$ variable induces a domain shift (gray hammer), whereas an intervention at the $X_e$ variable induces a covariate shift (black hammer).

We repeat in Figure 6 the causal graph for anomaly detection here for convenience. To prove Theorem 1, we use the following lemma.

**Lemma 2.** If $W$ and $E$ are confounded, then $X_a \perp E \mid W$. Otherwise, $X_a \perp E$.

*Proof.* We prove this via d-separation. We start with the case of $W$ and $E$ not being confounded. Note that there are two paths from $X_a$ to $E$. One via $W$ and the other via $Z$. The path via $W$ is blocked, because $X_e$ is a collision and neither $X_e$ nor its descendant $Z$ are observed. The path via $Z$

is blocked, because $Z$ is a collision with no descendants and is not observed. Hence, $X_a \perp E$, when $W$ and $E$ are not confounded. In the case of $W$ and $E$ being confounded, we can show analogously that $X_a \perp E \mid W$. Note that there are three paths from $X_a$ to $E$. The first one is via $U$, but this is blocked, because $W$ is in that path, it is not a collision, and it is observed. The second one is via $W$ and $X_e$, but $W$ is in that path and it is not a collision there, so that path is also blocked. The third one is via $Z$, but $Z$ is a collision in that path with no descendants and it is not observed, so it is also blocked. Since all paths are blocked, when $W$ is observed, we conclude that $X_a \perp E \mid W$, when $W$ and $E$ are confounded. $\square$

**Theorem 2.** Suppose that $f$ learns invariant representations. If $W$ and $E$ are confounded, then $Z \perp E \mid W$. Otherwise, $Z \perp E$.

*Proof.* Recall that if $f$ learns invariant representations, then we assume it to be $X_a$-measurable. This assumption is justified in Veitch et al. [2021]. As a result, since $Z = f(X_a, X_b)$, we have that $Z$ is $X_a$-measurable.

We assume that $W$ and $E$ are not confounded and show that $Z \perp E$ by proving that $p(Z \in A, E \in B) = p(Z \in A)p(E \in B)$, for any $A, B \in \mathcal{B}(\mathbb{R}^n)$. Note that $p(Z \in A, E \in B) = p(Z^{-1}(A) \cap E^{-1}(B)) = p(X_a^{-1}(C_A) \cap E^{-1}(B))$, for some Borel set $C_A$. The last equality follows from $Z$ being $X_a$-measurable, which implies that $Z^{-1}(A) = X_a^{-1}(C_A)$, for some Borel set $C_A$, by Definition 5. By Lemma 2, we have that $X_a \perp E$. This implies that $p(X_a^{-1}(C_A) \cap E^{-1}(B)) = p(X_a^{-1}(C_A)) p(E^{-1}(B)) = p(Z^{-1}(A)) p(E^{-1}(B)) = p(Z \in A)p(E \in B)$, which is what we wanted to show.

We now assume that $W$ and $E$ are confounded and show that $Z \perp E \mid W$ by proving that

$$p(Z \in A, E \in B \mid W \in C) = p(Z \in A \mid W \in C)p(E \in B \mid W \in C),$$

for any $A, B, C \in \mathcal{B}(\mathbb{R}^n)$. By an analogous argument, we can show that $p(Z \in A, E \in B \mid W \in C) = p(X_a^{-1}(C_A) \cap E^{-1}(B) \mid W^{-1}(C))$. By Lemma 2, we have that $p(X_a^{-1}(C_A) \mid W^{-1}(C)) p(E^{-1}(B) \mid W^{-1}(C))$. With arguments similar to those above, we can show that the last expression is equal to $p(Z \in A \mid W \in C)p(E \in B \mid W \in C)$, which is what we wanted to show. $\square$

# B  Dataset Details

## B.1  Camelyon17

Our realistic anomaly detection dataset was derived from the Camelyon17 dataset (Koh et al. [2021], Bandi et al. [2018]), and contains $3 \times 96 \times 96$ patches of whole-slide images of lymph node sections sourced from patients who may have metastatic breast cancer. This dataset encompasses tissue patches obtained from five different hospitals. The objective here is to accurately predict the presence of tumor tissue within the patches drawn from hospitals that were not part of the training data. Prior work has shown that differences in staining between hospitals are the primary source of variation in this dataset, however, other divergent factors in the sampling distribution include different acquisition protocols and patient populations (Tellez et al. [2019]).

The in-distribution data was comprised of $151,280$ images evenly distributed across three hospitals, or $100,810$ images evenly distributed across two hospitals depending on the training setting. The other out-of-distribution data covered two additional datasets, the first with $34,904$ patches, and the second with $85,054$ patches. Note that to adapt this dataset to the anomaly detection setting, only normal images were included in the in-distribution training data.

## B.2  Synthetic datasets

The synthetic datasets employed in this study were derived from the DiagViB-6 benchmark (Eulig et al. [2021]). This benchmark uniquely allows for the manipulation of five independent generative factors from colored images: overlaid texture, object size, object position, lightness, and saturation, in addition to the semantic features that correspond to the label. Our synthetic experiments utilized two datasets: MNIST (Deng [2012]) and Fashion-MNIST (Xiao et al. [2017]). All images in both datasets were upsampled to dimensions of $3 \times 256 \times 256$.

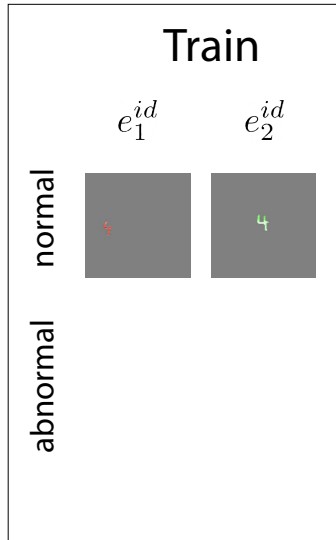 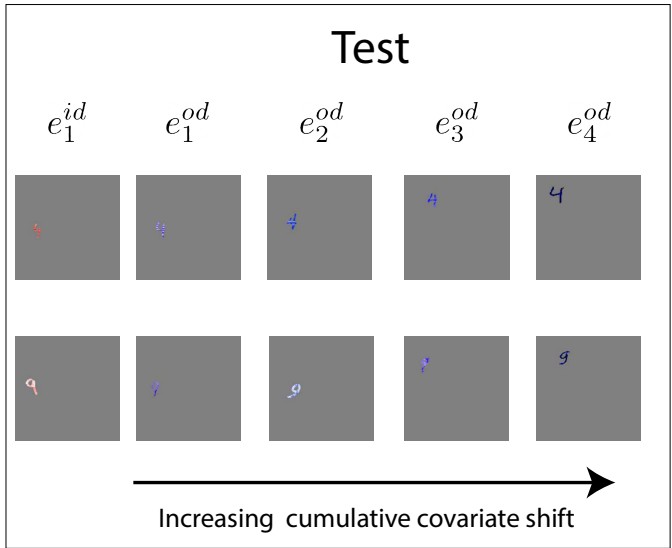

Figure 7: Illustration of our experimental setup for the synthetic covariate shift experiment. The image demonstrates representative examples of training data from two distinct environments, alongside instances of normal and abnormal test data subject to progressively accumulated covariate shifts. This configuration embodies the nuanced challenge of identifying subtle, yet potentially consequential, changes in the data distribution.

Initially, we generate two unique and distinct environments specifically designed for the training data. Our primary goal during this stage was to guarantee that all these factors exhibited noticeable differences when compared across the two generated environments.

Following the generation of these training environments, we proceeded to develop another pair of environments. These new environments were crafted for the validation data. To ensure consistency, these validation environments were fashioned in such a way that they closely mirrored or replicated the factor configuration that was present in the initial training environments, thus retaining an in-distribution setting.

In the final step, six additional environments, denoted as $e_0, e_1, ..., e_5$, were generated. Each environment $e_i$ consists of images in which $i$ factors have been altered with respect to $e_0$. For a depiction of the samples for these different environments, please refer to Fig. 7.

In devising our evaluation setup, we opted for inducing covariate shifts that are minor deviations from the original in-distribution environments. This decision was motivated by our goal to simulate subtle yet potentially detrimental covariate shifts, particularly in comparison to the challenge of differentiating normal from abnormal.

A description of the accumulated covariate shifts in the test environments, $e_0, e_1, e_2, e_3, e_4$, is provided in Table 1.

| $i$ | Chosen factor in $e_i$ |
|---|---|
| 0 | None |
| 1 | Hue |
| 2 | Texture |
| 3 | Lightness |
| 4 | Position |

Table 1: Environments $e_0, \ldots, e_4$ used in our synthetic benchmark. For $0 < i \leq 4$, the environment $e_i$ modifies the new factor indicated in the table in addition to the factors modified by $e_0, \ldots, e_{i-1}$.

## C   Invariantly pretrained encoders

We also extend our experimental evaluation by adding a comparison to Smeu et al. [2022], an environment-aware framework for AD that pretrains the encoder of the AD model using an invariance-inducing method (LISA or IRM). We evaluate this method in MNIST and F-MNIST subject to targeted covariate shifts (the exact same setup as seen in our original experiments). The method was incorporated into all baselines and compared to the same baselines while regularized through partial conditional invariance.

We show the results of these experiments in Fig. 8. Across all methods tested, and in both datasets, we observe a sharp decrease in performance when compared to a baseline non-invariant method and that it provides less robustness to covariate shifts than our proposed methodology.

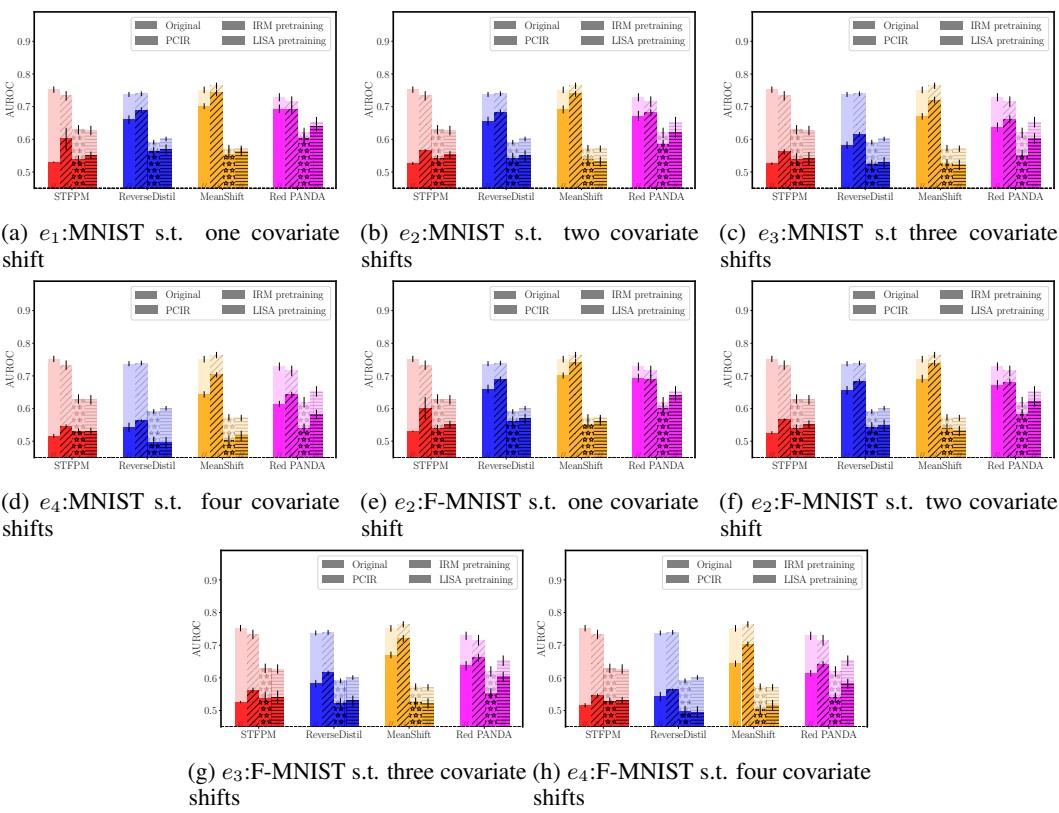

(a) $e_1$:MNIST s.t. one covariate shift

(b) $e_2$:MNIST s.t. two covariate shifts

(c) $e_3$:MNIST s.t three covariate shifts

(d) $e_4$:MNIST s.t. four covariate shifts

(e) $e_2$:F-MNIST s.t. one covariate shift

(f) $e_2$:F-MNIST s.t. two covariate shift

(g) $e_3$:F-MNIST s.t. three covariate shifts

(h) $e_4$:F-MNIST s.t. four covariate shifts

Figure 8: Experimental results MNIST and Fashion-MNIST with additional invariant pretraining following. (**background transparent bar-plots:** in-distribution evaluation; **foreground opaque bar-plots:** out-of-distribution evaluation). **(a-d)** Results in MNIST. **(e-h)** Results in Fashion-MNIST.

## D   Visualization of Invariance *vs* Informativeness

In Fig. 9 we plot the two-dimensional representation of the final layer of a model trained through MeanShift(Reiss and Hoshen [2021]) at different levels of PCIR regularization. The embeddings are obtained through t-SNE. From the progressive increase in the weight of the PCIR term, we see the increased superimposition of the different environments leading to more invariance at the loss of informativeness in the representation.

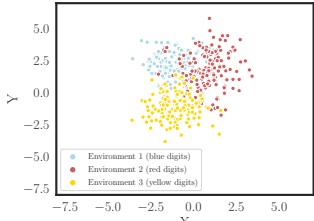
(a) Invariant and informative

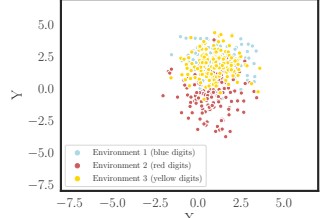
(b) Invariant but non-informative

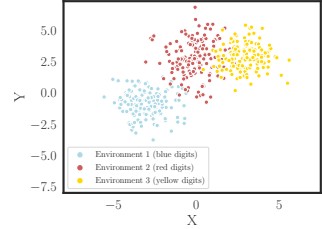
(c) Informative but non-invariant

Figure 9: TSNE embeddings of MNIST with three background colors for the digits 4 and 9. The model used was MeanShift subject to different degrees of partial conditional invariant regularization. **(a)** PCIR term set to 5 **(b)** PCIR term set to 150. **(c)** PCIR term set to 0.

# E   Tables of Results

| | **In dist.** | | **Out of dist.** | | |
| --- | --- | --- | --- | --- | --- |
| | $e_1$
($\uparrow$) AUROC | $e_2$
($\uparrow$) AUROC | $e_3$
($\uparrow$) AUROC | $e_4$
($\uparrow$) AUROC | $e_5$
($\uparrow$) AUROC |
| STFPM | $0.850 \pm 0.005$ | $0.883 \pm 0.014$ | $0.815 \pm 0.024$ | $0.753 \pm 0.007$ | $0.745 \pm 0.005$ |
| STFPM (PCIR) | $0.868 \pm 0.007$ | $0.873 \pm 0.008$ | $0.814 \pm 0.014$ | $0.798 \pm 0.011$ | $0.774 \pm 0.010$ |
| ReverseDistil | $0.853 \pm 0.009$ | $0.884 \pm 0.014$ | $0.797 \pm 0.015$ | $0.716 \pm 0.009$ | $0.723 \pm 0.004$ |
| ReverseDistil (PCIR) | $0.866 \pm 0.007$ | $0.903 \pm 0.007$ | $0.857 \pm 0.010$ | $0.744 \pm 0.021$ | $0.718 \pm 0.010$ |
| CFA | $0.625 \pm 0.002$ | $0.689 \pm 0.008$ | $0.566 \pm 0.014$ | $0.454 \pm 0.005$ | $0.590 \pm 0.017$ |
| CFA (PCIR) | $0.625 \pm 0.003$ | $0.700 \pm 0.007$ | $0.597 \pm 0.011$ | $0.473 \pm 0.009$ | $0.595 \pm 0.027$ |
| MeanShift | $0.751 \pm 0.012$ | $0.761 \pm 0.014$ | $0.731 \pm 0.015$ | $0.703 \pm 0.009$ | $0.701 \pm 0.010$ |
| MeanShift (PCIR) | $0.781 \pm 0.014$ | $0.791 \pm 0.014$ | $0.778 \pm 0.012$ | $0.772 \pm 0.013$ | $0.770 \pm 0.011$ |
| CSI | $0.601 \pm 0.021$ | $0.600 \pm 0.017$ | $0.582 \pm 0.014$ | $0.491 \pm 0.015$ | $0.531 \pm 0.014$ |
| CSI (PCIR) | $0.648 \pm 0.022$ | $0.652 \pm 0.021$ | $0.621 \pm 0.020$ | $0.581 \pm 0.018$ | $0.600 \pm 0.023$ |
| Red PANDA | $0.761 \pm 0.017$ | $0.764 \pm 0.015$ | $0.671 \pm 0.015$ | $0.641 \pm 0.015$ | $0.651 \pm 0.012$ |
| Red PANDA (PCIR) | $0.761 \pm 0.017$ | $0.769 \pm 0.015$ | $0.742 \pm 0.019$ | $0.731 \pm 0.015$ | $0.740 \pm 0.015$ |

Table 2: Experimental results on realist domain shift in the **Camelyon17** dataset, for both regularized and unregularized models trained on two environments. Results are presented over in-distribution evaluation (environments $e_1$ and $e_2$) and over out-of-distribution (environments $e_3$, $e_4$, and $e_5$).

| | **In dist.** | | | **Out of dist.** | |
| --- | --- | --- | --- | --- | --- |
| | $e_1$
($\uparrow$) AUROC | $e_2$
($\uparrow$) AUROC | $e_3$
($\uparrow$) AUROC | $e_4$
($\uparrow$) AUROC | $e_5$
($\uparrow$) AUROC |
| STFPM | $0.854 \pm 0.011$ | $0.873 \pm 0.004$ | $0.782 \pm 0.013$ | $0.771 \pm 0.013$ | $0.735 \pm 0.008$ |
| STFPM (PCIR) | $0.865 \pm 0.006$ | $0.873 \pm 0.005$ | $0.796 \pm 0.014$ | $0.782 \pm 0.012$ | $0.759 \pm 0.009$ |
| ReverseDistil | $0.830 \pm 0.028$ | $0.866 \pm 0.007$ | $0.781 \pm 0.004$ | $0.707 \pm 0.023$ | $0.710 \pm 0.033$ |
| ReverseDistil (PCIR) | $0.843 \pm 0.009$ | $0.880 \pm 0.015$ | $0.799 \pm 0.004$ | $0.714 \pm 0.010$ | $0.715 \pm 0.009$ |
| CFA | $0.667 \pm 0.004$ | $0.708 \pm 0.011$ | $0.628 \pm 0.019$ | $0.559 \pm 0.005$ | $0.643 \pm 0.006$ |
| CFA (PCIR) | $0.667 \pm 0.014$ | $0.705 \pm 0.019$ | $0.623 \pm 0.004$ | $0.561 \pm 0.003$ | $0.644 \pm 0.003$ |
| MeanShift | $0.742 \pm 0.014$ | $0.741 \pm 0.013$ | $0.681 \pm 0.014$ | $0.739 \pm 0.012$ | $0.690 \pm 0.013$ |
| MeanShift (PCIR) | $0.772 \pm 0.013$ | $0.784 \pm 0.012$ | $0.739 \pm 0.018$ | $0.747 \pm 0.013$ | $0.731 \pm 0.013$ |
| CSI | $0.620 \pm 0.015$ | $0.636 \pm 0.014$ | $0.548 \pm 0.013$ | $0.571 \pm 0.012$ | $0.541 \pm 0.015$ |
| CSI (PCIR) | $0.650 \pm 0.019$ | $0.650 \pm 0.010$ | $0.613 \pm 0.015$ | $0.624 \pm 0.014$ | $0.611 \pm 0.010$ |
| Red PANDA | $0.751 \pm 0.012$ | $0.742 \pm 0.013$ | $0.651 \pm 0.010$ | $0.751 \pm 0.010$ | $0.641 \pm 0.011$ |
| Red PANDA (PCIR) | $0.767 \pm 0.016$ | $0.781 \pm 0.014$ | $0.761 \pm 0.013$ | $0.770 \pm 0.012$ | $0.751 \pm 0.009$ |

Table 3: Experimental results on realist domain shift in the **Camelyon17** dataset, for both regularized and unregularized models trained on three environments. Results are presented over in-distribution evaluation (environments $e_1$, $e_2$, $e_3$) and over out-of-distribution (environments $e_4$, and $e_5$).

|  | In dist.
(↑) AUROC | Out of dist.
(↑) AUROC |
|---|---|---|
| STFPM | 0.699 ± 0.025 | 0.630 ± 0.025 |
| STFPM (PCIR) | 0.724 ± 0.020 | 0.698 ± 0.023 |
| ReverseDistil | 0.673 ± 0.057 | 0.617 ± 0.032 |
| ReverseDistil (PCIR) | 0.734 ± 0.013 | 0.723 ± 0.013 |
| CFA | 0.752 ± 0.003 | 0.705 ± 0.018 |
| CFA (PCIR) | 0.785 ± 0.003 | 0.759 ± 0.005 |
| MeanShift | 0.683 ± 0.041 | 0.629 ± 0.043 |
| MeanShift (PCIR) | 0.731 ± 0.022 | 0.726 ± 0.052 |
| CSI | 0.671 ± 0.026 | 0.626 ± 0.024 |
| CSI (PCIR) | 0.692 ± 0.017 | 0.674 ± 0.019 |
| Red PANDA | 0.732 ± 0.026 | 0.691 ± 0.031 |
| Red PANDA (PCIR) | 0.742 ± 0.029 | 0.721 ± 0.012 |

Table 4: Experimental results on realist shortcut learning in the **Waterbirds** dataset, for both regularized and unregularized. Results are presented over in-distribution evaluation and over out-of-distribution.

|  | In dist.
(↑) AUROC | 1 cov. shift
(↑) AUROC | 2 cov. shifts
(↑) AUROC | 3 cov. shifts
(↑) AUROC | 4 cov. shifts
(↑) AUROC |
|---|---|---|---|---|---|
| STFPM | 0.850 ± 0.005 | 0.883 ± 0.014 | 0.815 ± 0.024 | 0.753 ± 0.007 | 0.745 ± 0.005 |
| STFPM (PCIR) | 0.868 ± 0.007 | 0.873 ± 0.008 | 0.814 ± 0.014 | 0.798 ± 0.011 | 0.774 ± 0.010 |
| ReverseDistil | 0.853 ± 0.009 | 0.884 ± 0.014 | 0.797 ± 0.015 | 0.716 ± 0.009 | 0.723 ± 0.004 |
| ReverseDistil (PCIR) | 0.866 ± 0.007 | 0.903 ± 0.007 | 0.857 ± 0.010 | 0.744 ± 0.021 | 0.718 ± 0.010 |
| CFA | 0.625 ± 0.002 | 0.689 ± 0.008 | 0.566 ± 0.014 | 0.454 ± 0.005 | 0.590 ± 0.017 |
| CFA (PCIR) | 0.625 ± 0.003 | 0.700 ± 0.007 | 0.597 ± 0.011 | 0.473 ± 0.009 | 0.595 ± 0.027 |
| MeanShift | 0.751 ± 0.012 | 0.761 ± 0.014 | 0.731 ± 0.015 | 0.703 ± 0.009 | 0.701 ± 0.010 |
| MeanShift (PCIR) | 0.781 ± 0.014 | 0.791 ± 0.014 | 0.778 ± 0.012 | 0.772 ± 0.013 | 0.770 ± 0.011 |
| CSI | 0.601 ± 0.021 | 0.600 ± 0.017 | 0.582 ± 0.014 | 0.491 ± 0.015 | 0.531 ± 0.014 |
| CSI (PCIR) | 0.648 ± 0.022 | 0.652 ± 0.021 | 0.621 ± 0.020 | 0.581 ± 0.018 | 0.600 ± 0.023 |
| Red PANDA | 0.761 ± 0.017 | 0.764 ± 0.015 | 0.671 ± 0.015 | 0.641 ± 0.015 | 0.651 ± 0.012 |
| Red PANDA (PCIR) | 0.761 ± 0.017 | 0.769 ± 0.015 | 0.742 ± 0.019 | 0.731 ± 0.015 | 0.740 ± 0.015 |

Table 5: Experimental results on synthetic covariate shift over the **MNIST** dataset, for both regularized and unregularized models. Results are presented over in-distribution evaluation, and test sets subject to one to four different covariate shifts, as portrayed in Fig. 7

.

|  | In dist.
(↑) AUROC | 1 cov. shift
(↑) AUROC | 2 cov. shifts
(↑) AUROC | 3 cov. shifts
(↑) AUROC | 4 cov. shifts
(↑) AUROC |
|---|---|---|---|---|---|
| STFPM | 0.752 ± 0.010 | 0.530 ± 0.002 | 0.526 ± 0.004 | 0.526 ± 0.004 | 0.516 ± 0.007 |
| STFPM (PCIR) | 0.733 ± 0.014 | 0.601 ± 0.034 | 0.566 ± 0.004 | 0.562 ± 0.008 | 0.546 ± 0.006 |
| ReverseDistil | 0.737 ± 0.008 | 0.660 ± 0.013 | 0.655 ± 0.014 | 0.582 ± 0.011 | 0.543 ± 0.014 |
| ReverseDistil (PCIR) | 0.740 ± 0.007 | 0.689 ± 0.009 | 0.683 ± 0.008 | 0.615 ± 0.007 | 0.564 ± 0.004 |
| MeanShift | 0.751 ± 0.010 | 0.701 ± 0.009 | 0.692 ± 0.012 | 0.670 ± 0.010 | 0.643 ± 0.010 |
| MeanShift (PCIR) | 0.764 ± 0.010 | 0.742 ± 0.010 | 0.739 ± 0.009 | 0.720 ± 0.010 | 0.703 ± 0.008 |
| CSI | 0.691 ± 0.016 | 0.629 ± 0.011 | 0.571 ± 0.014 | 0.550 ± 0.012 | 0.542 ± 0.014 |
| CSI (PCIR) | 0.681 ± 0.014 | 0.661 ± 0.016 | 0.639 ± 0.014 | 0.611 ± 0.015 | 0.599 ± 0.011 |
| Red PANDA | 0.729 ± 0.013 | 0.693 ± 0.013 | 0.672 ± 0.015 | 0.637 ± 0.014 | 0.614 ± 0.010 |
| Red PANDA (PCIR) | 0.715 ± 0.016 | 0.690 ± 0.013 | 0.681 ± 0.010 | 0.661 ± 0.012 | 0.642 ± 0.009 |

Table 6: Experimental results on synthetic covariate shift over the **Fashion-MNIST** dataset, for both regularized and unregularized models. Results are presented over in-distribution evaluation, and test sets subject to one to four different covariate shifts, as portrayed in Fig. 7

# F    Performance Gained Compared to Baseline

Our investigation of the influence of partially conditional regularization on model performance is further expanded in Fig.10. This figure presents the percentile difference in mean-AUROC (Area Under the Receiver Operating Characteristic) between partially conditionally regularized and unregularized models.

By comparing each regularized model to its unregularized equivalent across all the tested environments, we have been able to observe a consistent improvement in performance when regularization is applied. This observation holds true across all models and out-of-distribution environments studied.

The increase in performance due to regularization varies from $2.5\%$ to as substantial as $20\%$ in some models. This variability implies a model-specific dependency on the degree of invariance that can be induced. Crucially, however, regularization substantially bolsters out-of-distribution performance without any additional training cost, reinforcing its merit as an effective strategy for developing robust anomaly detectors.

To further highlight this point and for the sack of completeness, we also present in Fig. 11, Fig. 12 and Fig. 13each regularized model compared to its baseline.

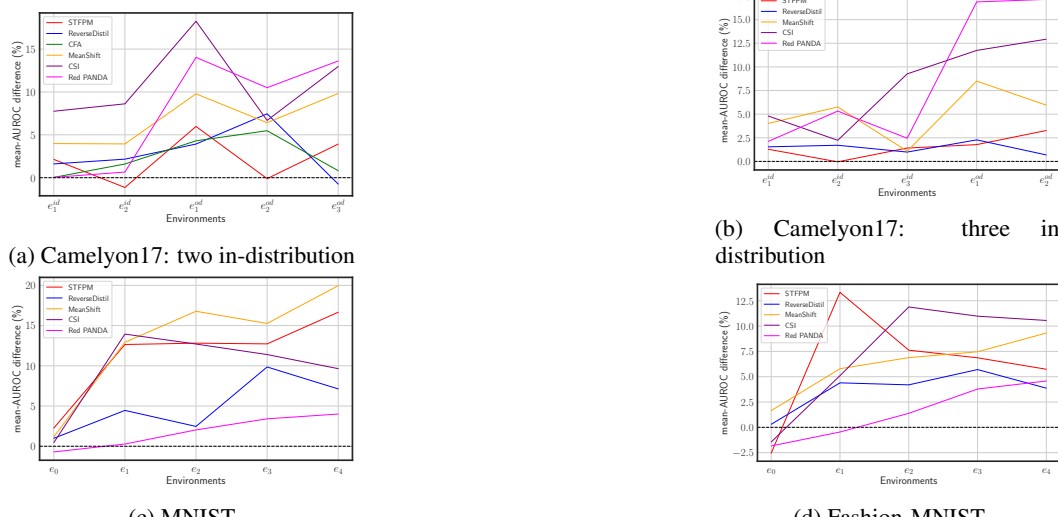

(a) Camelyon17: two in-distribution

(b)    Camelyon17:    three    in-distribution

(c) MNIST

(d) Fashion-MNIST

Figure 10: Percentual performance gain over each regularized model when compared to unreglarized baseline.

# G    Ablation Studies

As part of our comprehensive examination of how covariate shifts influence individual regularization weights, we have plotted the performance trajectories of all evaluated models, traversing environments from $e_0$ to $e_4$. These are captured in Fig.14 and Fig.15.

As expected, and already observed in both our main findings and previous work (Ming et al. [2022]), a review of these plots unveils a trend that permeates across all examined models: the performance of models appears to inversely correlate with the number of induced covariate shifts. As the complexity introduced by these shifts mounts, the models' performance experiences a proportionate and systematic decline. This observable trend is essentially monotonic, signifying a erosion in model performance with each incremental rise in the quantity of covariate shifts.

However, it is important to note that this trend is not without exceptions. In particular, when scrutinizing the data pertaining to environment $e_4$, we can observe anomalies to this downward trend. In these exceptional instances, despite the increase in the number of covariate shifts, the performance of certain models appears to resist the general declining pattern.

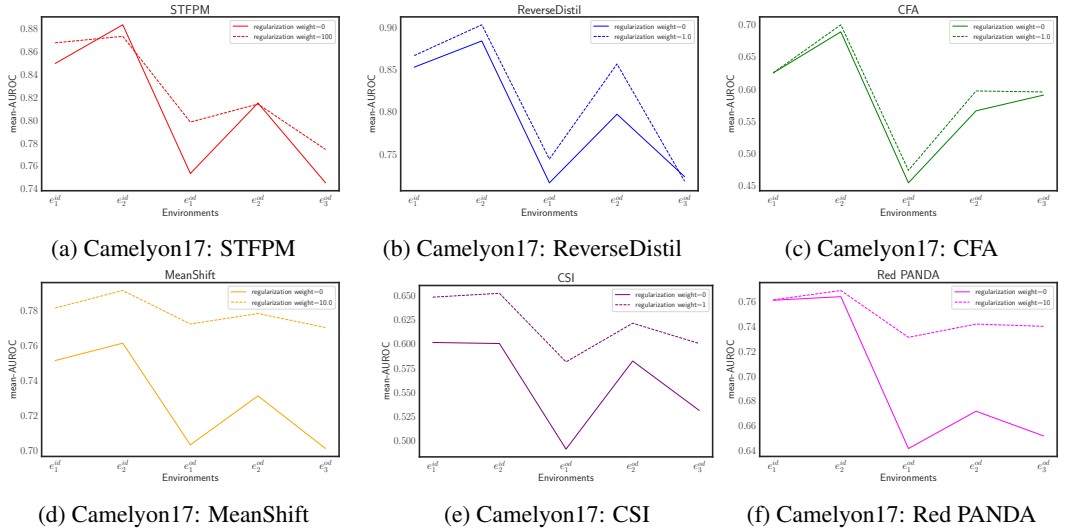

Figure 11: Mean-AUROC curve of each anomaly detector and its regularized version in the Camlyon17 dataset.

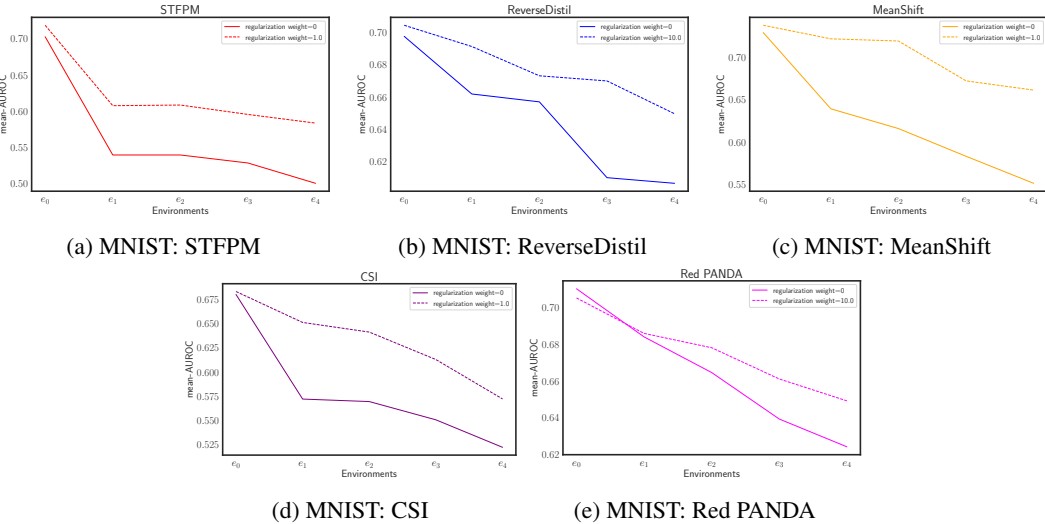

Figure 12: Mean-AUROC curve of each anomaly detector and its regularized version in the MNIST dataset.

Thus, our overall conclusion, while acknowledging these exceptions, is that the prevalence of covariate shifts largely contributes to a degradation in model performance.

It is however important to note that the unregularized methods still underperforms when compared to the same method under a even small amount (0.001) of partially conditional regularization added.

## H  Additional Discussion

### H.1  Shortcut Learning in Anomaly Detection

To expand on the experiment tackling real-world shortcut learning, and to better understand how a distribution shift affects different kinds of shortcut features (Geirhos et al. [2020]) captured by the model, we now will look at how inducing distinct changes to the anomaly detection causal graph may lead to malfunctions in the model.

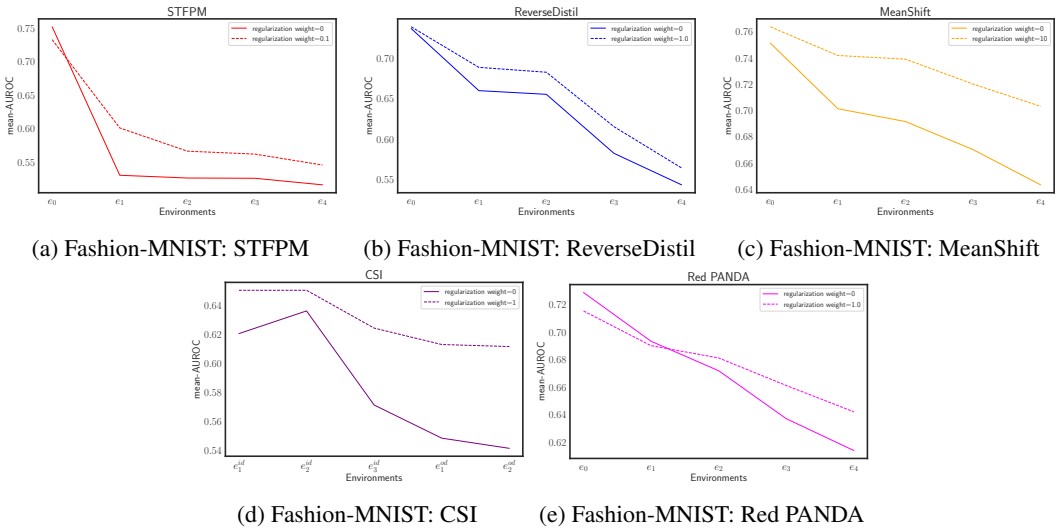

Figure 13: Mean-AUROC curve of each anomaly detector and its regularized version in the Fashion-MNIST dataset.

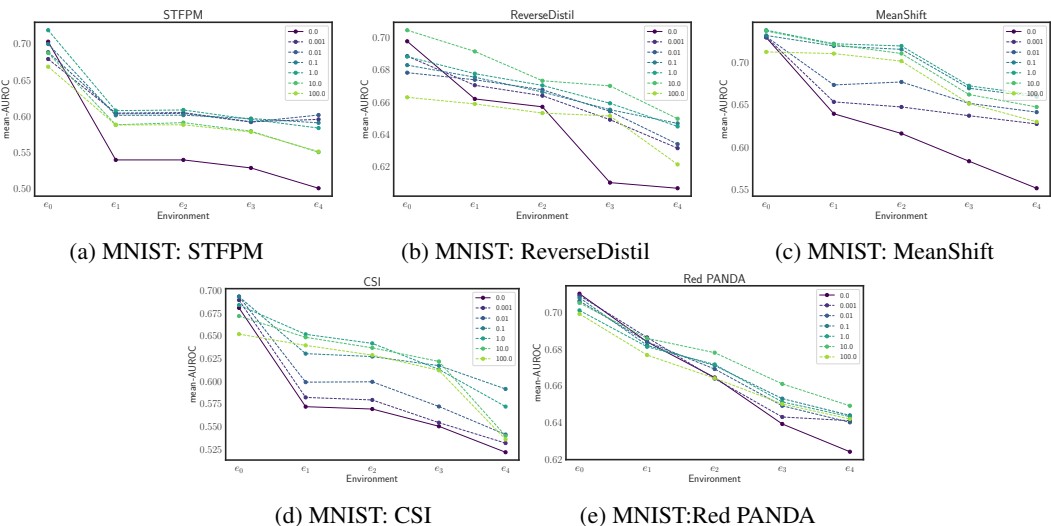

Figure 14: Ablation study over the weight of the regularization term for MNIST under distribution shift, with separate plots for each model.

Suppose we recover our formulation of the partition of an object $X$ into the semantic features that distinguish normal and abnormal samples, $X_a$, and into the style features induced by the environment, $X_e$. In that case, it is possible to distinguish between settings that may lead to a model failure when a shortcut feature is captured.

Let us simplify our analysis by considering the setting where the training data is sampled under the intervention $p^{do(E=e')}(X)$, that is the style features are fixed into a specific setting, $X'_e$. This is a prevalent setting in real-world applications as spurious correlations between style and semantic features may occur when sampling the training data. Remember that in the training set, $X_a$ only produces features of normal objects. Under this constraint, it was already previously noted that anomaly detection methods are particularly susceptible to capturing the style features as a prominent factor for the representation of $X$ (Ming et al. [2022]).

Moving to the evaluation stage of the anomaly detector, we can then consider two settings.

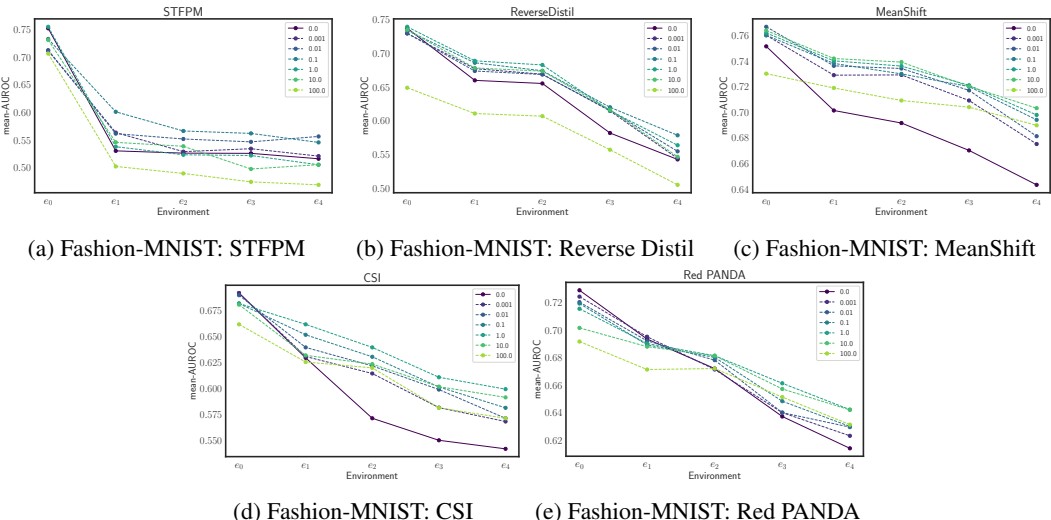

(a) Fashion-MNIST: STFPM     (b) Fashion-MNIST: Reverse Distil     (c) Fashion-MNIST: MeanShift

(d) Fashion-MNIST: CSI     (e) Fashion-MNIST: Red PANDA

Figure 15: Ablation study over the weight of the regularization term for Fashion-MNIST under distribution shift, with separate plots for each model.

The first setting consists of no changes to the intervention $p^{do(E=e')}(X)$, that is, there was no distribution shift in the sampling of the data. In this setting, the main surrogate for a model failure is derived from anomalies that are characterized by small changes in $X_a$ from normal to abnormal samples. That is, the style features would be considered the main source of information to characterize new samples, and under these constraints, it is only natural that all test instances would be highly likely to be set as normal. This setting was introduced by both Ming et al. [2022] and Cohen et al. [2023], and the underlying features can be referred to as *nuisance features*.

The second setting consists of a different intervention, $p^{do(E=e'')}(X)$ such that it differs from the training density $p^{do(E=e')}(X)$. In particular, we can consider an intervention that only changes stylistic features, $X_e$, captured by the model. We are essentially operating under a highly targeted covariate shift that focuses on the shortcut features. Therefore, depending on the extent of the changes in $X_a$ from normal to abnormal samples and how well they are captured by the model, this distribution shift would lead to an anomaly detector that classifies all new samples as anomalies.

Note that in both settings, as shown in our main presentation of the method, by inducing a partially conditional invariance to different environments, our regularization method also inherently introduces an invariance to the style features $X_e$. As supported by our results in the synthetic covariate shift experiments, we believe this also produces models that are not only robust to covariate shifts, as in the second scenario, but also to shortcut learning, as described in the first scenario.

### H.2 Data Augmentation

Although data augmentation is a well-established technique to tackle model robustness to distribution shifts, there are two main problems with solely relying on data augmentation to solve a problem of distribution shift. First, data augmentation relies on both knowing of the existence of a distribution shift, and a way to simulate it. This could be achievable when the distribution shift is characterized by transformations in easily identifiable attributes (e.g. the color of a digit), and which are also easy to simulate through data transformations. However, in real-world settings, this is rarely the case: distribution shifts are complex transformations that are almost impossible to identify, and in many cases impossible to programmatically simulate. For example, in the case of the Camelyon17 dataset that considers histological cuts from different hospitals, the changes between environments that are derived from the change in histological staining are easy to visually inspect, and could even be simulated. But, by changing the hospital, the patient population also changes, and with it several cofounders, such as group age or patient comorbidities. These are both impossible to know without further annotation and unfeasible to simulate. Furthermore, data augmentation does not induce an invariance to a particular distribution shift, giving no additional "guarantees". It only increases the

pool of examples of the data in hopes that the model implicitly captures the additional simulated variance in the images.

Yet, data augmentation could still be beneficial in the context of penalized invariant regularization, by producing meaningful augmentations in the context of a multi-environment setting. It could be used to generate additional data for a single intervention, thus increasing the pool of available samples of a specific environment, or be used to generate new data for a new intervention, leading to an increased number of environments available, and the overall pool of samples in the dataset. This would effectively alleviate the second drawback of solely using data augmentations.

### H.3    Choice of Distance Metric

We considered other options aside from MMD, like the Wasserstein distance. MMD is known to be computationally easier than Wasserstein's distance. This is because MMD can be easily computed using Gaussian kernels, whereas the Wasserstein distance requires computing a supremum over a set of probability measures, which is computationally intractable in general.

Another option we considered was the KL-divergence. However, previous works have demonstrated that this divergence is very unstable for probability distributions supported on low-dimensional manifolds (Arjovsky et al. [2017]). Note that many of the methods used for anomaly detection and machine learning are trained on samples from such distributions. A similar argument applies for variations of this divergence like Shannon-Jensen divergence and total variation.

Finally, we remark that MMD has a strong theoretical foundation. In spite of its simplicity, it is derived from a supremum of differences of expected values of functions from a reproducible-kernel Hilbert space (Gretton et al. [2012]). It has been shown that when this metric is 0 then the two distributions must match, which is precisely what we aim to achieve when learning representations that are invariant to the environments.

## I    Implementation Details

### I.1    Baselines

**STFPM**    The STFPM (Wang et al. [2021]) algorithm incorporates a pretrained teacher network and a student network that share the same structure. The student network assimilates the distribution of images devoid of anomalies by aligning the features with corresponding features in the teacher network. To boost robustness, the algorithm uses multi-scale feature matching. This multi-tiered feature alignment lets the student network absorb a blend of multi-level insights from the feature pyramid, thereby permitting the detection of anomalies of varying magnitudes.

During the inference stage, the feature pyramids of both teacher and student networks are put into comparison. A larger discrepancy between these pyramids implies a heightened likelihood of the presence of anomalies.

**Reverse distillation**    The reverse distillation method (Deng and Li [2022]) is assembled from three networks: an initial pretrained feature extractor, $f$, a bottleneck embedding, $\phi$, and the student decoder network, $\nu$. The primary layer, or the backbone, of $f$ is derived from a *ResNet* model that was pretrained on the ImageNet dataset.

During the execution of a forward pass, the model extracts features from three separate ResNet blocks. These features are encoded by amalgamating the three feature maps using the multi-scale feature fusion block of $\phi$, and then transferred to the decoder, $\nu$, which is constructed to mirror the feature extractor, albeit with operations reversed.

Throughout the training process, the output of these mirrored blocks is made to match the outputs from the respective layers of the feature extractor. This is ensured by adopting the cosine distance as the loss metric.

**CFA**    Feature Adaptation based on CFA (Lee et al. [2022]) identifies anomalies utilizing features that are specifically tailored to the target dataset. The CFA model comprises two main elements: firstly, a learnable patch descriptor that learns and assimilates features oriented towards the target,

and secondly, a scalable memory bank that remains unimpacted by the size of the target dataset. In conjunction with a pre-trained encoder, CFA applies a patch descriptor and memory bank. By doing so, it makes use of transfer learning to bolster the density of normal features. Consequently, this facilitates an easier distinction between normal and abnormal features.

**Mean-shifted**   The mean-shifted contrastive learning method (Reiss and Hoshen [2021]) introduces a novel loss function that calculates angular distances using the mean of all feature vectors as a reference point. This is done in contrast to using the origin as a reference and the Euclidean distance. It also combines two loss functions, one involving contrastive terms akin to Chen et al. [2020], but these terms are positioned around a hypersphere centred on the mean of all feature vectors. To deter positive samples from repelling themselves, it also incorporates an angular centre loss that encourages samples to gravitate towards the mean of normal samples.

**CSI**   CSI (Tack et al. [2020]) is a direct extension of Chen et al. [2020], introducing a unique form of data augmentations known as distribution-shifting augmentations. In this setup, distribution-shifted augmentations are treated as negative samples instead of positive ones and are consequently pushed away from all positive samples. These augmentations include manipulations such as rotations and permutations. The augmentation's potential to shift the distribution is assessed through the AUROC, where samples altered by the said augmentation are considered out-of-distribution samples. The underlying notion here is that distinguishability is directly proportional to the shift in distribution.

**Red PANDA**   The Red PANDA method for anomaly detection Cohen et al. [2023] tackles the particular problem of anomaly detection under nuisance or distracting features. Relying on labels from the nuisance factors, it employs a contrastive disentanglement loss following Kahana and Hoshen [2022], in conjunction with a perceptual loss to train a generator function end-to-end with a pretrained encoder.

## I.2   Anomaly Scoring

**STFPM**   During training, the student feature tries to align the distribution of training dataset with the teacher. During prediction, the input $x$ is fed into both student and teacher feature extractors, where the student outputs $f_s(x)$ and teacher outputs $f_t(x)$. For the anomaly scoring, it relies in a traditional density estimation approach. The assumption would be that the normal samples are mapped to the high density area where the student encoder and the teacher encoder are aligned, and the anomalous samples are mapped to the low density area of the student extractor. Therefore, the anomaly score is computed as the distance between $f_s(x)$ and $f_t(x)$, i.e. $AS(x) = d(f_s(x), f_t(x))$. In this case the distance metric $d$ is the $l_2$-norm.

**Reverse distillation**   The anomaly scoring function here is derived from the standard anomaly scoring functions used in reconstruction-based anomaly detection algorithm. In particular, the anomaly score is defined as the distance between the encoded features and the reconstructed features from the decoder. $AS(y) = d(f(y), \nu(\phi(f(y)))$, where $\nu(\cdot)$ is the decoder, $\phi(\cdot)$ is the distiller and $f(\cdot)$ is the encoder. The idea behind this anomaly scoring function is that the decoder has learned to reconstruct normal samples due to training dataset, but isn't able to reconstruct anomalous samples. Therefore, anomalous samples would have a higher anomaly score.

**CFA**   For CFA we use the standard image-level density estimation approach. In particular, the training samples are mapped to a feature map $f(x)$ and clustered into $k$ clusters using $k$-means. For the prediction, given an input $y$, its final feature $f(y)$ is computed after feeding it into the feature extractor and descriptor. Then the $d$ nearest cluster centers of $f(y)$ would be selected, and the anomaly score for that sample is the mean distance to those $d$ centers. $AS(y) = \Sigma_{f_i \in N_k(x)} d(f_i, f(y))$. In this case, the distance metric $d$ is simply the $l_2$ distance. The idea behind this approach is to assume that the anomalous samples would be mapped to low-density area in feature space, which are far away from all the cluster centers, while the normal samples would be mapped to high density area in feature space, which is close to the normal samples in training dataset.

**Mean-shift and Red PANDA**   Both contrastive learning based methods used as baselines, namely, mean-shift (Reiss and Hoshen [2021]) and Red Panda (Cohen et al. [2023]) rely on finetuning an

encoder (both pre-trained or not) by grouping the set of feature vectors from images in the training data around a sub-region of the hypersphere centered in the origin. During prediction time, the most common approach to classify a new sample as anomaly or not is through the mean distance of the $k$NN normal images. Following the original work, in mean-shift, $k$ was set to 2, and in Red Panda it was set to 1.

**CSI**   CSI (Tack et al. [2020]) relied on only a vanilla contrastive loss between original samples, and highly augmented samples to serve as a proxy for abnormal samples. Similarly to the previous setting of mean-shift and Red Panda, this leads to a feature space that falls around the hypersphere centered in the origin, but not necessarily in the surface. However, operating with the underlying hypothesis that the highly augmented samples match the anomalies, the feature vectors from images in the training data are already being pushed to the diametrically opposite side of the hypersphere when compared to abnormal samples. Additionally, it was empirically verified that the norm of vectors of abnormal samples is much lower than that of in-distribution samples. This leads to a distance criterion that measures the closest training sample through the cosine similarity and the norm of the feature vector of the sample: $\max_m \text{sim}(f(x_m), f(x)) \cdot |f(x)|$, where $f$ is the encoder that maps the input object, $x$, to its feature space, and sim is the cosine similarity.

### I.3   Hyperparameters

As this work considered a novel setting where each anomaly detection method was evaluated for the first time, we modestly optimized hyperparameters. Our approach consisted of two primary steps. The first involved scaling up two key factors: (a) batch size, and (b) learning rate. Subsequently, we methodically scanned through an array of distinct parameters for each baseline model. These included the backbones ResNet18, ResNet34, ResNet50 and WideResNet50, alongside various anomaly scoring methodologies that leverage image-level, density estimation, reconstruction error, and pixel-wise density estimation approaches. An additional aspect of our study was an ablation analysis where the regularization weight was fine-tuned by sweeping through the set of values $0.001, 0.01, 0.1, 1, 10, 100$. We adhered to the hyperparameters as depicted in the original works for all other variables and refrained from performing any further optimizations on them.

### I.4   Backbone choice

One thing we notice during our experiments is that for models that rely on the pretrained backbones, the choice of backbone matters. For instance, for STFPM, the optimal choice was the simplest feature extractor ResNet18, but for reverse distillation, the optimal choice was WideResNet50. The choices seem to be model dependent more than dataset relevant. For more details on the chosen backbone for each method refer to Tab. 7

### I.5   Computational resources

The complete project required 3400 hours of GPU usage throughout all experiments, covering development, testing, and comparisons. The resources supplied were part of a local custer, and consited of two GPU models: the NVIDIA TITAN RTX and the NVIDIA Tesla V100.

### I.6   Code and Licensing

The main Python libraries used in our implementation, were Pytorch, which is under a BSD-3 license[1], and Pytorch Lightning, which is under Apache 2.0 license[2].

Methods that were derived from the anomalib library (Akcay et al. [2022]), namely STFPM, reverse distillation, and CFA, were already implemented as a Pytorch Lightning Module, and are all under an Apache 2.0 license[3]. These were incorporated directly in our pipeline.

---

[1]https://github.com/pytorch/pytorch/blob/main/LICENSE
[2]https://github.com/Lightning-AI/lightning/blob/master/LICENSE
[3]https://github.com/openvinotoolkit/anomalib/blob/main/LICENSE

|  | STFPM | ReverseDistil | CFA | MeanShift | CSI | Red PANDA |
|---|---|---|---|---|---|---|
|  | Wang et al. [2021] | Deng and Li [2022] | Lee et al. [2022] | Reiss and Hoshen [2021] | Tack et al. [2020] | Cohen et al. [2023] |
| **Camelyon17** (3x96x96) | | | | | | |
| learning rate | $10^{-2}$ | $10^{-2}$ | $10^{-5}$ | $10^{-3}$ | $10^{-3}$ | $10^{-4}$ |
| optimizer | SGD | Adam | AdamW | Adam | Adam | SGD |
| batch size | 32 | 32 | 16 | 32 | 64 | 128 |
| backbone | ResNet18 | WideResNet50 | ResNet18 | ResNet50 | ResNet18 | ResNet50 |
| pretaining | True | True | True | True | False | True |
| best reg. weight | 100 | 10 | 100 | 10 | 1 | 10 |
| anomaly score | density estimation | reconstruction error | density estimation | density estimation | density estimation | density estimation |
| **DiagViB-6 (MNIST)** (3x256x256) | | | | | | |
| learning rate | $10^{-1}$ | $10^{-2}$ | - | $10^{-3}$ | $10^{-3}$ | $3 \cdot 10^{-4}$ |
| optimizer | SGD | Adam | - | Adam | Adam | SGD |
| batch size | 32 | 32 | - | 32 | 32 | 32 |
| backbone | ResNet18 | WideResnet50 | - | ResNet50 | ResNet18 | ResNet50 |
| pretaining | True | True | - | True | False | True |
| best reg. weight | 1 | 10 | - | 1 | 1 | 10 |
| anomaly score | density estimation | reconstruction error | - | density estimation | density estimation | density estimation |
| **DiagViB-6 (Fashion-MNIST)** (3x256x256) | | | | | | |
| learning rate | $10^{-1}$ | $10^{-2}$ | - | $10^{-3}$ | $10^{-3}$ | $3 \cdot 10^{-4}$ |
| optimizer | SGD | Adam | - | Adam | Adam | SGD |
| batch size | 32 | 32 | - | 32 | 32 | 32 |
| backbone | ResNet18 | WideRestNet50 | - | ResNet50 | ResNet18 | ResNet50 |
| pretraining | True | True | - | True | False | True |
| best reg. weight | 0.1 | 1 | - | 10 | 1 | 1 |
| anomaly score | density estimation | reconstruction error | - | density estimation | density estimation | density estimation |

Table 7: A detailed summary of the hyperparameters used for each evaluated model across three datasets: Camelyon17, DiagViB-6 (MNIST), and DiagViB-6 (Fashion-MNIST). Parameters include learning rate, scheduler, optimizer, batch size, backbone, pretraining, regularization weight, and mmd kernel type, along with the type of anomaly score. Notably, the CFA model could not be successfully implemented for DiagViB-6 based experiments despite trying an extensive range of hyperparameter combinations. Models are referenced by their respective citations.

DCoDR (Kahana and Hoshen [2022]), which Red Panda is based from, was released under a Software Research License[4]. Our experiments for Red Panda were derived directly from the official repository. Mean-shifted contrastive learning was released under a Software Research License[5]. We re-implemented this method as a Pytorch Lightning Module, loosely following its original official implementation. DiagViB-6 (Eulig et al. [2021]) and the Camelyon17 (Koh et al. [2021]) datasets were also publicly released with a GNU Affero General Public License v3.0[6] and a MIT License[7], respectively. Our implementation follows directly from its official repository.

---

[4]https://github.com/jonkahana/DCoDR/blob/main/LICENSE

[5]https://github.com/talreiss/Mean-Shifted-Anomaly-Detection/blob/main/LICENSE

[6]https://github.com/boschresearch/diagvib-6/blob/main/LICENSE

[7]https://github.com/p-lambda/wilds/blob/main/LICENSE

