# OpenReview forum: "Invariant Anomaly Detection under Distribution Shifts: A Causal Perspective"
_NeurIPS.cc/2023/Conference — NeurIPS 2023 poster_

### Official Review · Reviewer_AUeF · 2023-07-04

**Soundness:** 3 good
**Presentation:** 1 poor
**Contribution:** 3 good
**Rating:** 4
**Confidence:** 3

**Summary:**

This paper tackles the problem of anomaly detection under a distribution shift. The authors describe the setting in detail, where the training is drawn from multiple subsets of examples, and test data is drawn from different subsets. The subset of each training example is known. The goal is to detect anomalies which are not subset specific. The authors then present a regularization term that aids finding anomalies which generalize over different subsets.

**Strengths:**

1.	The tackled problem is important for real-life applications (as presented by the authors).
2.	The presented method is simple to implements and shows a notable and consistent improvement on top of all other methods.


**Weaknesses:**

Overall I find the idea interesting, yet I have 3 main concerns regarding the paper: (i) Writing quality. (ii) Setting and mathematical formulation novelty. (iii) Benchmarked datasets.

1)	Setting and Mathematical Formulation: In the paper the authors claim to present a novel setting for anomaly detection as a key contribution (contribution #2, lines 70-71). Yet in fact, a similar setting was presented in Red PANDA when treating a nuisance feature as a sub-domain. Moreover, the mathematical formulation is almost identical to the one presented in DCoDR [], which Red PANDA is built over, but no attribution is mentioned. I believe claiming novelty on the presented setting is unjustified. Also, using the formulation used in Red PANDA can save some of notations presented, which complicated the writing.
2)	Writing Quality: I believe the paper is not written well, most noticeable the experimental section (Section 5). It is hard to follow to exact experimental design of the authors, and some key parts are missing in its description: (i) The experimental design of the synthetic benchmarks is unclear about how many subsets appear in the training data (important for baselines such as Red PANDA). (ii) The results are presented in bar charts, making it impossible to get an exact evaluation. (iii) Each method is evaluated by 4 different bar chart results. I didn’t see any part in the paper (or the supplementary materials) that clearly explains what each result describes. (iv) The proposed method is an addition of a regularization term to other existing AD methods. While the term itself is described clearly, the adaptation of other AD methods is not.
3)	Datasets Selection: Another concern regards the dataset used for the evaluation in the paper. I believe the method should be tested in more challenging scenarios, as presented in Red PANDA. The Edges2Shoes and SmallNORB datasets should be much more challenging than MNIST and Fashion-MNIST.

I have also some other minor concerns:
1)	In the ablation study of the regularization weight (lines 273-282) states “This pattern suggests a simple linear search could be sufficient to identify an optimal weight for the regularization term”. I believe this statement is ill-posed, as this is only true if given some labelled anomalies for constructing the test set. In real-life scenarios, this is not always the case.
2)	Minor Mistakes:
a.	Lines 131-132: all definition refer to f_1.
b.	I believe the notation of E in line 192 should be X_e.


[1] Kahana, Jonathan, and Yedid Hoshen. "A contrastive objective for learning disentangled representations." European Conference on Computer Vision. Cham: Springer Nature Switzerland, 2022.

**Questions:**



**Limitations:**

1.	The setting requires a labelling of the domains / nuisance attribute during training.

---

> ### Author Rebuttal · Authors · 2023-08-10
>
> We thank the reviewer for their time and thorough review of our work, as well as the comments and proposed suggestions.
>
> **1. (i) Yet in fact, a similar setting was presented in Red PANDA when treating a nuisance feature as a sub-domain.**
> We agree with the reviewer that nuisance features when accompanied by a particular intervention to a covariate is part of our setting. In particular, a nuisance feature can be treated as a shortcut feature, i.e. a stylistic feature that could help the model to perform well in-distribution but fails in out-of-distribution generalization. We provide further analysis of shortcut features and of their vulnerability to covariate shifts in Section 3 of the Appendix.
>
> **Changes:** We will update our manuscript by describing the connection of how our work tackles shortcut features, to the analysis of nuisance features as described by Red Panda.
>
> **1. (ii) Moreover, the mathematical formulation is almost identical to the one presented in DCoDR [], which Red PANDA is built over, but no attribution is mentioned.**
>
> We thank the reviewer for bringing to our attention the connection between our work and the work in disentangled representation learning, DCoDR.
>
> Although we acknowledge that DCoDR poses a double requirement of invariant and informative representations, its mathematical formulation of invariance is different from ours and entails an alternative take on invariant representation learning. In particular, it defines the invariance condition as $p(E|Z) = p(E)$, where $E$ denotes the environment and $Z$ is the learned representation. It then imposes a separate condition *per domain/environment*, such that $P(Z|E)=U_s$, where $U_s$ is a uniform spherical distribution, which is enforced through a contrastive loss term. On the other end, using the *do* notation, we formulate our invariance condition through a pairwise equality across environments, as $p^{do(E=e)}(Z)=p^{do(E=e')}(Z)$. This formulation ultimately leads to our partial conditional distribution invariance, a distinct condition from the one formulated in DCoDR.
>
> Furthermore, we highlight that contribution #2 covers robustness to distribution shifts, both covariate and domain shifts, and more importantly, the mathematical formulation of an anomaly detection problem under causal inference - which neither DCoDR nor Red PANDA do.
>
> Therefore, we argue that contribution #2, covering both setting and mathematical formulation of distribution shifts through the lens of causal inference, is a novel contribution to the field of AD.
>
> **Changes:** We will reference DCoDR as part of our related work (Section 2.2.) as an example of a recently proposed and effective invariant representation learning method.
>
> **2. (i) The experimental design of the synthetic benchmarks is unclear about how many subsets appear in the training data.**
> Although the number of environments included in the training data (two) of the synthetic covariate shift experiments can be found in line 23 of the Appendix (as well as further details on the data generation process), we thank the reviewer for pointing out that this detail should also be in the main manuscript.
>
> **Changes:** We have updated lines 236-237 as follows:
> *"The training data is comprised of two distinct environments, generated by manipulating original instantiations of each factor, and ensuring all factors differed between the two environments. For the test data, we defined five distinct environments denoted as $e_0, e_1, ..., e_4$. The environment $e_0$ corresponds to images from MNIST for a specific digit under an original instantiation of each factor from one of the two training environments."*
>
> **2. (ii) The results are presented in bar charts, making it impossible to get an exact evaluation.**
> We agree with the reviewer on the need to provide exact values for all experimental results. We chose to present the results in bar charts for better interpretation and comparison between methods.
>
> **Changes:** We will add all the results in table format to the Appendix.
>
> **2. (iii) Each method is evaluated by 4 different bar chart results. I didn’t see any part in the paper (or the supplementary materials) that clearly explains what each result describes**
> To better visualize the drop in performance from in-distribution to out-of-distribution, we have overlaid the bar plots. In all figures the in-distribution setting corresponds to the transparent bar plot, and the out-of-distribution corresponds to the fully opaque bar plot. This is information was provided only in the caption of Figure 3.
>
> **Changes:** We will update the captions of Figures 3 and 4 with this additional information (see caption of results for the waterbirds experiment), and add these details to the text of Section 5.2.
>
> **2. (iv) The proposed method is an addition of a regularization term to other existing AD methods. While the term itself is described clearly, the adaptation of other AD methods is not**
>
> As covering the details of each adapted method would span too much content in the main manuscript, we have provided all implementation details for all methods in Section 5 of the Appendix, including hyperparameters, anomaly scoring, and backbone choice.
>
> **3. I believe the method should be tested in more challenging scenarios (...).The Edges2Shoes and SmallNORB datasets should be much more challenging than MNIST and F-MNIST.**
>
> We agree with the reviewer's assessment of MNIST and F-MNIST. We would however point out that, besides these two datasets, we have also evaluated all methods in Camelyon17. This is a challenging and widely used benchmark for domain generalization, in a real-world medical imaging setting.
> Moreover, we have added now experiments on an additional dataset with real-world natural images: the Waterbirds dataset (see global comment). In these, we also demonstrate the shortcomings of all baseline methods (including Red PANDA) and the benefits of introducing our regularization term.

---

> > ### Comment · Reviewer_AUeF · 2023-08-20
> >
> > 1.i-ii) Setting is similar to Red PANDA / DCoDR:
> > I agree with the authors that their formulation is a generalization of the formulation presented in DCoDR, since the covariance shift is not mentioned in Red PANDA or DCoDR.
> > Yet, I still believe the settings are very similar, as the domain can be treated as a nuisance feature, in the Red PANDA view. I believe the manuscript (Sections 2.2, 3) should be heavily modified to address that, without claiming contribution about “…the dual requirements of invariance and informativeness for robust AD models in the face of distribution shifts” (lines 70-71, contribution #2).
> >
> > 2 i-iii) Writing Quality: I believe the authors have carefully addressed my concerns in these part.
> >
> > 2 iv) Adaptation: There might be a misunderstanding here: what I meant is that the authors would state that their method is an adaptation (in the form of an additional objective) to previous methods. This is a minor concern. Moreover, I believe this should be presented as an advantage of the proposed method, as it makes it slightly more robust.
> >
> > 3) More Challenging Scenarios: I believe the additional external dataset satisfies my concerns. However, given the relation to Red PANDA, I think it is important to evaluate on its proposed datasets (SmallNorb and Edges2shoes) as well, since these benchmarks were proposed for the same task.
> >
> > General Comment:
> > The authors addressed several of my concerns. In addition I like the general idea of the paper and its results seem promising. Saying that, I believe that this manuscript is not yet ready. Many changes have to be done (attribution to previous art in settings definition and changes to the evaluation section) as mentioned above. For these reasons, I keep my grading the same.

---

> > > ### Author Response · Authors · 2023-08-21
> > > **Reply to Reviewer AUeF**
> > >
> > > We thank the reviewer for reading our rebuttal and providing additional feedback on our work.
> > >
> > > 1. To clarify our previous response (point 1. ii), our formulation is not a generalization of DCoDR/RedPanda.
> > > In particular, DCoDR is tackling disentangled representation learning, whereas our work was formulated for anomaly detection. Both approaches aim to address invariant representation learning, yet they follow divergent methodologies and principles, which lead also to two different formulations of invariance.  Our formulation allowed us to derive *Corollary 1* and the principle of *partial conditional invariance*, which would not be feasible under the DCoDR framework.
> > >
> > > More specifically, the formulation of invariance in DCoDR is $p(E|Z)=p(E)$, and has led to a single environment-wise invariance objective, where image pairs are drawn from the same environment. This is different from our formulation of an invariance criteria. We define invariance as $p^{do(E=e)}(Z) = p^{do(E=e')}(Z)$. This led to a pairwise, multi-environment constraint, where images are drawn from pairs of environments and compared through our partial conditional invariant regularization.
> > >
> > > We will acknowledge DCoDR in the related work as an example of a recently proposed and effective invariant representation learning method, which relies on contrastive learning.
> > >
> > > Furthermore, treating domains as nuisance features, as proposed by the reviewer, would negate our ability to derive both Corollary 1 and the principle of partial conditional invariance. It was our formalization with causal inference what led to these insights, which are essential for effectively detecting anomalies under distribution shifts.
> > >
> > > Finally,  we understand the importance of clear articulation of our contributions. We are not claiming novelty over the proposal of a double requirement of invariance and informativeness, but instead the formulation of these concepts through causal inference. We will modify contribution #2 to:
> > > *“We use causal inference to propose a novel formalization of the well-established dual requirements of informativeness and invariance for robust AD models, leading to a new insight, as described in Corollary 1.”*
> > >
> > > 2. (iv) We thank the reviewer for this observation and agree that it makes our contribution more robust.
> > >
> > > 3. We recognize that these datasets were indeed evaluated in the RedPanda work. However, it's worth noting that this is the only anomaly detection work where these specific datasets have been used. Similarly, to FashionMNIST and MNIST, they can be viewed as toy datasets tailored for niche scenarios. They also involve nuisance features rather than the broader domain or covariate shifts.
> > >
> > > In our research, we have already tested our methods on synthetic datasets such as MNIST and FashionMNIST, which serve similar purposes of understanding the method's performance in controlled settings. To ensure a comprehensive evaluation, we extended our experiments to more realistic and complex datasets, namely Camelyon17 and Waterbirds. We believe these datasets present a diverse range of challenges, making them suitable benchmarks for our approach.
> > >
> > >
> > > **Final comment:**
> > > In closing, we are grateful for the reviewer's recognition of the paper's innovative approach and our empirical evaluations showcasing promising results. We have made deliberate efforts to elucidate the distinction and potential overlaps between our work and DCoDR/RedPanda. By explicitly acknowledging DCoDR in the related work and clarifying our unique formulations and contributions, we aspire to dispel any ambiguities. We hope that these targeted revisions address your concerns.

---

> ### Comment · Area_Chair_WBMr · 2023-08-19
> **Request for Reviewer AUeF to respond to authors' comments**
>
> Reviewer AUeF, as the discussion period is nearly over, would you please read the authors' response and explain the extent to which their answers address your concerns, and whether you will adjust your rating.
>
> If you decide to keep your score, please justify this decision in detail, specifying which aspects of the paper or response have been the deciding factors in you keeping your score.

---

### Official Review · Reviewer_yiE8 · 2023-07-04

**Soundness:** 3 good
**Presentation:** 3 good
**Contribution:** 3 good
**Rating:** 6
**Confidence:** 4

**Summary:**

The paper focuses on the making anomaly detection models robust to various kinds of distribution shifts over normal/benign samples. This goal is achieved by formalizing the anomaly detection problem from the information theory perspective and introducing the requirements it brings for developing AD models that are robust to distribution shifts. Based on this observation, the paper proposes a regularization term that ensures the partial distribution invariance across environments. Lastly, the authors have shown the improved robustness of their proposed method on numerous problems.

**Strengths:**

The paper provides an original approach to improve the robustness of anomaly detection models to distribution and covariant shifts on the data. The mutual information based penalty on the latent space is a very creative idea. The authors did a really good job on explaining the problem and motivating their work especially at Section 3.2. Moreover, the analysis of drawbacks of current AD methods such as STFM, Reverse Distillation etc, leaves no questions marks on the purpose of this work and helps readers to easily navigate through the paper. I also found Figure 1 very explanatory (even though I have small question).

**Weaknesses:**

As I have discussed in the strength part, I mainly liked the organization of the paper and the authors did a good job on motivating their work. I had hard time to understand the purpose of Figure 2. I would also suggest authors to provide more details on the proof of Corollay 1 and include more explanation on the notations. For the sake of the flow, I would also suggest authors to include the definition of the technical terms such as d-separation, confounding, together wit the notations.

Lastly, I am slightly concerned about the limitations of this method in practice. I am not too much familiar with other AD problems that suits the problem setup given in the section 3.1.; therefore, I could not think of problems other than MNIST and FMINST that this method can be implemented to. I would be very happy if authors can comment on that.

**Questions:**

- Could you elaborate more on why Figure 1 (d) is not invariant?
- Could you provide more details on the proof of Corollary 1? Why conditioning on W would make the paths blocked?
- What are the advantages or disadvantages of the penalized problem over data augmentation?
- The MNIST and Fashion-MNIST datasets are good sources to do the comparisons and rather easy examples to demonstrate the distribution and covariant shifts. I was wondering if the authors can provide few more sophisticated examples to demonstrate this problem? Considering the limited time of the authors I would be very satisfied with a brief discussion on additional one or two more datasets.


**Limitations:**

The authors successfully discussed about some of the limitations of the method in the paper.

---

> ### Author Rebuttal · Authors · 2023-08-09
>
> We thank the reviewer for the positive evaluation, constructive feedback, and thorough review of our work. We will address the pointed weaknesses jointly with raised questions.
>
> **I had hard time to understand the purpose of Figure 2.**
> The goal of Figure 2, was to help visualise the formulation given in Section 3, but also provide additional insights on the causal structure of the AD problem that is essential for the derivation of Corollary 1.
>
> **Changes:** We have expanded Figure 2 with Z now being presented in the graphic, and will refer to it in Corollary 1 for its independence assumption.
>
> **On the invariance of Figure 1 (d).**
>  We thank the reviewer for overall positive feedback on Figure 1.
> Regarding the posed question, through our formulation, in the context of AD, we can only guarantee invariance is attained within the support of p* (W=0). Invariant representation of each p_do(E=e_i) with respect to the environment variable, would mean that normal samples (depicted as the circles) are being mapped to the same region of $Z$ irrespectively of the intervention to the variable E. In (d), an contrarily to (b), we observe that the distributions green, blue, and red, do not overlap inside the support of p* (in yellow).
>
> For an empirical example, we have added a TSNE plot of the embeddings of a deep anomaly detector under different levels of regularization (see global response).
>
> **Could you provide more details on the proof of Corollary 1?**
> We have added a more detailed proof of Corollary 1. In particular, we explain why all paths from Z to X_e are blocked. We give in appendix a recap on d-separation and confounding. We make it clearer that when f learns invariant representations, f must be X_a-measurable, and therefore, X_e cannot influence Z, which removes the arrow from X_e to Z in the diagrams. This shows then that any path from Z to X_e is blocked.
>
> **Why conditioning on W would make the paths blocked?**
> Conditioning on W makes W observed. Note that any path that goes from Z to E goes either through W or through X_e. Paths through X_e are blocked, because X_e does not influence Z, as Z = f(X) is X_a measurable. Paths through W are blocked, by d-separation, because W is observed and in none of those paths, the arrows are all pointing to W.
>
> **What are the advantages or disadvantages of the penalized problem over data augmentation?**
> This has is an interesting question that we have also posed ourselves.
>
> Although data augmentation is a well-established technique to rely on in the context of model robustness to distribution shifts, there are two main problems with solely relying on data augmentation to solve a problem of distribution shift. First, data augmentation relies on both knowing of the existence of a distribution shift, and a way to simulate it. This could be achievable when the distribution shift is characterized by transformations in easily identifiable attributes (e.g. the color of a digit), and which are also easy to simulate through data transformations. However, in real-world settings, this is rarely the case: distribution shifts are complex transformations that are almost impossible to identify, and in many cases impossible to programmatically simulate. For example, in the case of the Camelyon17 dataset that considers histological cuts from different hospitals, the changes between environments that are derived from the change in histological staining are easy to visually inspect, and could even be simulated. But, by changing the hospital, the patient population also changes, and with it several cofounders, such as group age or patient comorbidities, that are both impossible to know without further annotation and unfeasible to simulate. Furthermore, data augmentation does not induce an invariance to a particular distribution shift, giving no additional “guarantees”. It only increases the pool of examples of the data in hopes that the model implicitly captures the additional simulated variance in the images.
>
> Yet, data augmentation could still be beneficial in the context of penalized invariant regularization, by producing meaningful augmentations in the context of a multi-environment setting. It could be used to generate additional data for a single intervention (increasing the pool of available samples of a specific environment), or be used to generate new data for a new intervention (increasing the number of environments available, and the overall pool of samples in the dataset). This would effectively alleviate the second drawback of solely using data augmentations.
>
> **I was wondering if the authors can provide few more sophisticated examples to demonstrate this problem?**
> We appreciate the reviewer's request for a more realistic evaluation setting.
> First, we would like to highlight that we already include in our work experiments in Camelyon17. This is a real-world dataset of medical images where we tackle AD for predicting the existence of tumorous lesions in histological cuts, whilst subject to a domain shift derived from a change of hospital. Although, as mentioned in the previous point, the precise nature of the features changed when intervening in hospital variable, by employing our partial conditional distribution regularization we are able to induce some invariance to the distribution shifts, as demonstrated through our improvements of 2 to 8% in the o.o.d AUROC.
>
> Also, we have included another real-world natural image dataset: the Waterbirds dataset. This is a realistic dataset where the distribution shift is induced through changes in the habitat depicted in the image background. Similarly to Camelyon17, we also demonstrate an improvement in the robustness of the AD models when accompanied by our proposed regularization term. For more details on this experiment, we refer the reviewer to the global comment.

---

> > ### Comment · Reviewer_yiE8 · 2023-08-12
> >
> > I would like to thank authors for their details answers and clarifying my concerns. I think the additional evidence they have provided addresses my concerns. I would also suggest adding discussion on the data augmentation authors made here into the paper or supplementary material which could help strengthen the paper.

---

> > > ### Author Response · Authors · 2023-08-12
> > > **Response to Reviewer yiE8**
> > >
> > > We would like to thank the reviewer for the response and update to the rating.
> > > Thank you for your constructive feedback and for recognizing the efforts we made in our rebuttal. We are pleased to know that our explanations and additional evidence were effective in addressing your concerns.
> > >
> > > We appreciate your suggestion regarding the inclusion of the data augmentation discussion in the manuscript. We believe this will indeed enhance the quality and clarity of our work, and we will ensure to incorporate it accordingly.

---

### Official Review · Reviewer_ZicU · 2023-07-06

**Soundness:** 2 fair
**Presentation:** 3 good
**Contribution:** 2 fair
**Rating:** 5
**Confidence:** 3

**Summary:**

This paper aims to promote AD through the lens of causality. To this end, the authors propose to minimize the distribution discrepancy of the leaned representations among environments. Some experiments are conducted to verify the effectiveness of the proposed method.

**Strengths:**

1 Exploring AD through a causal perspective is promising.

2 One of the main contribution of this work is the empirical observation, where the authors show that existing AD method may fail when meeting distribution shifts. Meanwhile, the experimental results are solid and the performance gain is decent.

**Weaknesses:**

1 The motivation figure is vivid, while it is built upon synthetic data. I suggest the authors to give a corresponding figure using features learned with the proposed method, which may be a direct support to the efficacy of the proposed method.

2 The definitions used in this work is confusing, where the authors use both (content, style) and (related, style) notation. A clear definition is required.

3 Corollary 1 is confusing. The authors claim Z=f(X), however, f is typically trained with the data X. Consequently, the independent mechanism assumption does not hold any more. Thus, Z should not appear in Corollary 1.

4 The novelty of the proposed method is limited, where the authors merely control the MMD for two distributions with different representations, i.e., varying with environments.

Typos:
Line 131, f1, f1, and f1 should be f1, f2, and f3.

**Questions:**

cf Weaknesses

**Limitations:**

cf Weaknesses

---

> ### Author Rebuttal · Authors · 2023-08-09
>
> We thank the reviewer for their comments and suggestions to improve this work.
>
> **1. I suggest the authors to give a corresponding figure using features learned with the proposed method.**
> We agree with the reviewer's suggestion and have produced a TSNE plot of representations produced with different levels of regularization.
> Such a figure is presented in Figure 3 of the .pdf attached to the global answer.
>
> **Changes:**
> We will add this figure to our manuscript.
>
> **2. The definitions used in this work is confusing, where the authors use both (content, style) and (related, style) notation. A clear definition is required.**
> We have tried to find in our work the usage of "related" notation, but failed to do so. Could the reviewer point us to a specific example where it was used. We would appreciate the chance to clarify any confusion in our notation.
>
> **3. Corollary 1 is confusing. The authors claim Z=f(X), however, f is typically trained with the data X. Consequently, the independent mechanism assumption does not hold any more. Thus, Z should not appear in Corollary 1.**
>
> We thank the reviewer for carefully reviewing our proof.
>
> This is correct: $f$ is trained with $X$, so $Z$ is, in general, not independent of $X$. However, we assume that f learns invariant representations. Under this assumption, we demonstrate Corollary 1. We exploit the insight that f must be $X_a$-measurable, which means that $X_e$ does not influence the outcome of $f$. This observation allows us to demonstrate via d-separation that $Z$ is independent from $E$, when f learns invariant representations.
>
> **Changes:**
> We changed the proof of Corollary 1 to highlight that $f$ is trained with data $X$. We have also added to the Appendix a more detailed proof. In particular, we explain why all paths from $Z$ to $X_e$ are blocked and added a recap on d-separation and confounding.
>
> **4. The novelty of the proposed method is limited, where the authors merely control the MMD for two distributions with different representations.**
>
> In this section, we would like to expand on the novelty and value of the methodological contribution and would like to make the case that our method can actually be instrumental within the context of AD subject to distribution shifts.
>
> In particular, matching the two distributions from different environments is a essential condition for generating invariance towards, for example, a covariate shift. Solving this problem would then be akin to solving domain generalization under this induced shift. (if the invariance across distributions translates to new domains). To our best understanding, this is still an unresolved problem in the field of machine learning, and the most recent attempts have been led by causal approaches (as we describe in our related work in Section 2.2.) Furthermore, approaching a invariant representation learning using conditional distribution invariance is non-trivial, due to the nature of AD only revealing normal samples during training.
>
> MMD has been used before to regularize the training of models, but what makes this insight valuable is that MMD itself is not enough. It requires the conditioning on W = 0.  Likewise, we believe that having a solid theoretical grounding for our method and using causality is a much better approach as it not only provides theoretical foundations (c.f. Corollary 1), but also enhances our understanding of the problem and opens the door for further improvements.
>
> A final important observation is that in spite of being a conditional MMD, and in spite of being only a necessary condition, it is surprisingly strong enough to greatly improve the state-of-the-art anomaly detectors in the presence of domain shifts.
>
> We see the use of MMD as only one of the results of our causal analysis on anomaly detection. Our analysis has additionally provided the insights above on anomaly detection in the presence of distribution shifts, setting the stage for additional advancements relying on causal inference tools.

---

> > ### Comment · Reviewer_ZicU · 2023-08-20
> > **Thank you for your response.**
> >
> > Although the proposed methodology has been explored in our fields, it is among the pioneer in studying OOD detection from the causal perspective. The authors address my concerns, and I would like to raise my score.

---

> > > ### Author Response · Authors · 2023-08-20
> > > **Response to reviewer ZicU**
> > >
> > > We would like to thank the reviewer for updating the rating. We are glad to see that our clarifications and information effectively addressed your concerns.

---

> ### Comment · Area_Chair_WBMr · 2023-08-19
> **Request for reviewer Reviewer ZicU to respond to author's comments**
>
> Reviewer ZicU, as the discussion period is nearly over, would you please read the authors' response and explain the extent to which their answers address your concerns, and whether you will adjust your rating.
>
> If you decide to keep your score, please justify this decision in detail, specifying which aspects of the paper or response have been the deciding factors in you keeping your score.

---

### Official Review · Reviewer_ePdN · 2023-07-07

**Soundness:** 3 good
**Presentation:** 2 fair
**Contribution:** 2 fair
**Rating:** 4
**Confidence:** 3

**Summary:**

The paper tackles the problem of anomaly detection under distribution shifts from the perspective of causal inference. Causal inference aims to induce invariance across environments. The authors derive a regularization term for anomaly detection to achieve this goal. The resulting detector is robust to domain and covariate shifts, improving the detection performance under specific distribution shifts. Experiments are conducted on one real medical image datasets involving domain shifts and one synthetic dataset based on transforming MNIST.

**Strengths:**

* Authors experiment on a real-world dataset Camelyon17, which composes of distribution shifts, to demonstrate the effectiveness of their method.
* I like the way of motivating the problem (Fig. 1).
* As far as I know, this work is the first to apply causal inference techniques to anomaly detection tasks (which could have many citations). However, as a work that sets up the stage, the authors can do more rather than mere applications (see Weakness).

**Weaknesses:**

As the first work that tries to set up the stage for invariant anomaly detection with causal inference, I think there are still some todo's.
* There is a lack of justification of using the proposed solution rather than other possible causal learning techniques, e.g., invariant risk minimization or other methods that avoid shortcut learning. Unaddressed justification questions like: What is the uniqueness of the proposed solution compared to other causal learning methods? Why do you choose MMD instead of other discrepancy metrics? What are the potential adverse consequences of conditioning on only normal samples (in the case of W=0) and why do we not need to worry about? etc.
* I'm not convinced about the definitions of covariate shifts and domain shifts. In the current formulation, I don't see the difference between Def. 1 and Def. 2 because line 117 states: "the environment E only influences $X_e$". A clarification or concrete examples to illustrate the definitions could be helpful.
* Experiments are kind of similar and lack variability to demonstrate the significance of the proposed method. The environment only change the number of shifts at training time or at test time. There are some known image datasets in shortcut learning literature (waterbirds and CelebA) and OOD detection literature (CIFAR10-C). Maybe showcasing the proposed method in those datasets as well is more persuasive as they are dedicated to this setup and also real word natural images.

**Questions:**

Here is a mixed list of questions and suggestions:
* It seems Def. 3 (mutual information) and Section 3.4 are kind of redundant.
* Why not put variable $Z$ into Fig. 2? In this way, the independence assumption can be readily read off from the figure.
* In Figure 3 and Figure 4, which bar corresponds to in-distribution performance and which one corresponds to out-of-distribution performance?

**Limitations:**

The method requires multiple-domain training data, which could be infeasible under some circumstances.

---

> ### Author Rebuttal · Authors · 2023-08-09
>
> We thank the reviewer for their time and constructive feedback.
>
> **1. (A) There is a lack of justification of using the proposed solution.**
> We agree with the reviewer that our work would benefit from a more comprehensive background of other causal methods.
>
> We verified that the state-of-the-art in AD is based on deep representation learning, which sets the stage for invariant representation learning methods. Other techniques like invariant risk minimization or FISH [A] could not be adequate, as they are optimization techniques that were not designed for learning invariant representations. Furthermore, as we have pointed out in Section 2.2 (lines 104-105), it has been shown that IRM underperforms in real-world settings.
>
> For other methods, such as LISA [B], we see that they only tackle specific types of distribution shifts through selective data augmentation, and therefore require an understanding of the type of shift that has been applied to the data.  This can be prohibitive in many realistic settings.
>
> **Changes:**
> For our next version, we have produced the following concrete changes. First, we summarize the discussion above and incorporate it into our introduction while motivating. With this, the reader can see why we formalize anomaly detection using causality theory and choose to use follow an invariant representation learning approach. In addition, this shows why other causal learning techniques like invariant risk minimization or methods against shortcut learning are not enough to solve this problem. Second, we expand our related work section 2.2 by explaining why other shortcut learning-avoiding methods are not suited for the constraints of anomaly detection in the presence of domain shifts.
>
> [A] Shi, Yuge, et al. "Gradient matching for domain generalization." ICLR 2022.
> [B] Yao, Huaxiu, et al. "Improving out-of-distribution robustness via selective augmentation." ICML 2022
>
> **1. (B) Why MMD instead of other discrepancy metrics?**
> We considered other options aside from MMD, like the Wasserstein distance. MMD is known to be computationally easier than Wasserstein’s distance. This is because MMD can be easily computed using Gaussian kernels, whereas the Wasserstein distance requires computing a supremum over a set of probability measures, which is computationally intractable in general.
>
> Another option we considered was the KL-divergence. However, previous works have demonstrated that this divergence is very unstable for probability distributions supported on low-dimensional manifolds [C]. Note that many of the methods used for anomaly detection and machine learning are trained on samples from such distributions. A similar argument applies for variations of this divergence like Shannon-Jensen divergence and total variation.
>
> Finally, we remark that MMD has a strong theoretical foundation. In spite of its simplicity, it is derived from a supremum of differences of expected values of functions from a reproducible-kernel Hilbert space [D]. It has been shown that when this metric is 0 then the two distributions must match, which is precisely what we aim to achieve when learning representations that are invariant to the environments.
>
> **Changes:**
> Following this discussion, we have added a section to our Appendix motivating the use of MMD where we highlight the reasoning behind our choice.
> [C] Arjovsky, Martin et al "Wasserstein generative adversarial networks."ICML 2017
> [D] Gretton, Arthur, et al. "A kernel two-sample test." JMLR 2012
>
> **1. (C) Consequences of conditioning on only normal samples**
> Please see the global answer.
>
> **2. In the current formulation, I don't see the difference between Def. 1 and Def. 2.**
> Indeed, environment E only influences X_e, but interventions on X_e may be different from interventions on E. For example, consider intervening only on certain pixels of an image. This would be an intervention on X_e, but not an intervention on E, as the environment has not been modified to another valid environment. This shows also that interventions on E are stronger than interventions on X_e. For this reason, we focus on this work on methods that can detect anomalies even in the presence of these stronger interventions.
>
> **Changes**
> We added this remark to Section 3.3.
>
> **3. Experiments are kind of similar and lack variability to demonstrate the significance of the method.**
> Following the reviewer's suggestion, we have added additional experiments in Waterbirds. See global answer for more details.
>
> We would also like to point out that the synthetic covariate shift experiments not only change the number of shifts at training and test time but also evaluates the model with unseen style instantiation, as described in Section 5.1, lines 236-241. For example, the model is trained with red and purple digits positioned in the upper left and right corner of the image, and evaluated with yellow digits in the lower central position. Furthermore, the experiments in Camelyon17 serve a distinct purpose of evaluating domain generalization under a real-world medical imaging setting, with unknown environment interventions.
>
> **Further comments:**
> - We will summarize Section 3.4 and merge it into the previous section.
> - We will add Z to Figure 2
> - In all figures, the in-distribution setting corresponds to the transparent bar plot, and the out-of-distribution corresponds to the fully opaque bar plot. We provided this information for Figure 4, but not for Figure 3. To make it clearer, we will update both figures with the caption of the Waterbirds experiment.
>
> **Limitation**
> We would like to point out that the referenced limitation is acknowledged in our discussion (2nd paragraph of Section 6), where we provide possible solutions. These include relying on additional meta-data, which in many real-world settings is available, to overcome non-existence of domain labels [E].
> [E] Y. Lin et al. "When and how to learn invariance without environment partition?" NeurIPS 2022

---

> > ### Comment · Reviewer_ePdN · 2023-08-19
> >
> > Thank you for your answers. I'm still concerned about some aspects of the work. To the justifications, the current paper applies existing techniques in invariant representation learning to the AD setup and observes improvement when spurious correlation exists. Could you validate why other techniques couldn't lead to such improvement in AD setup? What is unique of your method to anomaly detection except conditioning on W=0? In addition, why is IRM not designed to learn an invariant representation? Without addressing these questions, I don't see the significant contribution to the AD area except verifying some techniques from other general problems can be applied for AD when the same problem present.

---

> > > ### Author Response · Authors · 2023-08-20
> > > **Reply to Reviewer ePdN**
> > >
> > > We thank the reviewer for the additional questions and comments.
> > >
> > > **1. Could you validate why other techniques couldn't lead to such improvement in AD setup?**
> > > As part of our rebuttal, we have actually extended our experimental evaluation to incorporate a comparison with other invariant representation learning methods (please refer to 2. in the general comment and additional *.pdf* for more details, and note that we promise to include these results in the main paper).
> > >
> > > In particular, we have shown that leveraging IRM or LISA to achieve invariance to covariate shifts is non-trivial in an anomaly detection setting. Our experiments show that not only do these methods underperform compared to a baseline non-invariant method but also offer less robustness against covariate shifts than our proposed partial conditional invariant regularization.
> > >
> > > Furthermore, RedPanda, one of the methods set as a baseline in our experiments, was also proposed to induce invariant representations in AD. RedPanda relies on inducing invariance as $P(Z|E)=U_s$, where $U_S$ is a uniform spherical distribution enforced single environment-wise. In contrast to our approach that induces invariance pairwise across multiple environments.  However, as our experiments demonstrate, contrastive learning-based methods are not enough, as RedPanda is still vulnerable to both covariate and domain shifts.
> > >
> > > Overall, our observations suggest that the performance limitations of prior methods in the AD domain might be attributed to unaddressed theoretical nuances. Our work endeavored to fill these gaps, and we believe that this approach has been instrumental in achieving the observed improvements.
> > >
> > >
> > > **2. On the uniqueness of the method**
> > > We agree with the reviewer that the paper builds upon existing work on representation learning. In particular, on [Veitch et al, 2022] which induces conditions for invariance across domains, and in particular, in classification problems where there are representative samples for each class are available. Our setting was not considered before, as applying such techniques to anomaly detection is non-trivial, and the theoretical foundations for invariant representation learning do not apply to anomaly detection, as there is no representative sample of anomalies while training.
> > > Therefore, the uniqueness of our contribution is tied to the formalization of anomaly detection using causal inference, leading to Corollary 1, whose proof is non-trivial. In fact, we were asked to provide more detailed explanations of this proof throughout the reviewing process. Indeed, it is this theoretical grounding that justifies a partial conditional invariance on W=0.
> > >
> > > Although the conditioning on W=0 is simple, it required an involved formalization and a non-trivial mathematical justification. Furthermore, this simple idea is powerful enough to provide large improvements in performance w.r.t. the state of the art in several different out-of-distribution settings, as our experiments demonstrate. These include spurious correlations, but also much more complex distributions shifts, as evidenced by our benchmark in Camelyon17.
> > >
> > > **3. In addition, why is IRM not designed to learn an invariant representation?**
> > > Indeed Invariant Risk Minimization was designed to learn an optimal classifier consistent across multiple training distributions. However, it's important to distinguish between the invariance of a classifier and the invariance of the learned underlying data representation. While IRM ensures that a classifier built on top of a data representation remains consistent across different environments, it doesn't necessarily guarantee that the data representation itself is invariant. The objective of IRM primarily focuses on the classifier's performance across various distributions rather than on the properties of the representation directly.
> > > We believe this justification to be partially the reason for the poor performance of IRM in our (additional) experiments in 2. of the global rebuttal comment.
> > >
> > > We hope to have addressed and clarified any remaining questions or uncertainties the reviewer may have regarding the contributions of our work.

---

> ### Comment · Area_Chair_WBMr · 2023-08-19
> **Request for reviewer Reviewer ePdN to respond to author's comments**
>
> Reviewer ePdN, as the discussion period is nearly over, would you please read the authors' response and explain the extent to which their answers address your concerns, and whether you will adjust your rating.
>
> If you decide to keep your score, please justify this decision in detail, specifying which aspects of the paper or response have been the deciding factors in you keeping your score.

---

### Official Review · Reviewer_zVDG · 2023-07-07

**Soundness:** 3 good
**Presentation:** 3 good
**Contribution:** 2 fair
**Rating:** 7
**Confidence:** 4

**Summary:**

The paper proposes a formalism for detecting anomalies in the presence of domain and covariate shifts. Data X is assumed of being decomposable into X_a and X_e where the latent factors X_a determine if X is an anomaly while X_e represents style features. To obtain invariant features, a regulariser is proposed. It is based on MMD between the features of samples from different environments, conditioned that they represent normal samples. Experiments on synthetic and real data show that the proposed regulariser improves the performance of multiple anomaly detection methods.

**Strengths:**

1. The paper is clearly written and the formalization is sound.

 2. The MMD regularisation is appropriate for the given graph assumptions.

 3. Experiments show good results in multiple settings.

**Weaknesses:**

1. A similar framework for anomaly detection in the presence of style changes is proposed in [A]. They also aim to solve anomaly detection based on content features (similar to X_a) while being invariant to style features (similar to X_e). Similarities and differences between the frameworks should be discussed.

2. Conditional MMD methods have been applied before for domain generalization [B,C]. This is completely fine, is shown that the proposed baseline is appropriate learning invariant features. The authors should discuss the relation to previous works in domain generalization and highlight the importance of partial conditioning.

3. It is not clear that partial conditioning on W = 0 (apply MMD only on normal data) is enough for obtaining invariant representations in the confounded case. It doesn’t seem the case. The paper should have a discussion about this aspect.
It is noted that “domain shifts are stronger than covariate shifts” but this is not formalized anywhere nor is it considered in the method. How important is this fact for the overall framework?

4. Some clarifications should be made about the following: The description in the background section (3.1) does not seem to match the causal graphs in Figure 2. “Conceptually, X_a represents the component of X that determines whether X is an anomaly, while X_e comprises style features. These style features, while unaffected by the anomaly status of an object, are influenced by the environment.” This description would suggest that the anomaly status (W) will affect only X_a but not X_e, but we have an arrow between W and X_e in Figure 2. Can the authors clarify this?


[A] Smeu et al. "Env-aware anomaly detection: Ignore style changes, stay true to content!." arXiv (2022)
[B] Long et al. “Transfer feature learning with joint distribution adaptation,” ICCV (2013)
[C] Kang et al. “Contrastive adaptation network for unsupervised domain adaptation,” CVPR (2019)

**Questions:**

See Weaknesses.

**Limitations:**

There is no negative societal impact.

---

> ### Author Rebuttal · Authors · 2023-08-09
>
> We thank the reviewer for the positive evaluation of the paper, and their comments and suggestions for improvement.
>
> **1. A similar framework for anomaly detection in the presence of style changes is proposed in [Smeu et al. 2022].**
>
> We thank the reviewer for identifying this framework. Although this work also proposes an environment-based approach for robust AD, no connection is made to covariate or domain shifts, nor to any causal inference study in AD. It proposes a two-step method: first, an encoder is pretrained using an environment-based approach (e.g. IRM) in an unrelated pretext task using the same data, second, an AD method is trained on top of the encoder. This both increases the cost of training such a model and deprives the possibility of using strong pre-trained models, which SOTA AD are highly reliant on. To evaluate this framework we have incorporated it into the baselines (state-of-the-art DL-based AD methods) and evaluated its performance in the MNIST and Fashion MNIST under covariate shifts. Our experiments demonstrate that the approach both underperforms when compared to a baseline non-invariant method and provides less robustness to covariate shifts than our proposed partial conditional invariant regularization. For more details please refer to the global response, point 2.
>
> **Changes**: We will add these results to our experimental section and will update our related work section accordingly.
>
> **2. The authors should discuss the relation to previous works in domain generalization and highlight the importance of partial conditioning.**
>
> We agree with the reviewer that the additional discussion and references provide increased validation for our approach.
>
> **Changes:**
> We have therefore extended the introduction with the following sentences :
> *"Similiar approaches relying on MMD regularization for domain generalization in different tasks have proven to be valuable [B, C]. However, to our knowledge, all methods relying on conditional invariance have been applied under the assumption of access to all classes during training. This is not the case in the AD setting."*
>
> We also updated line 212  in section 4.1 as follows:
>  *“We call this approach partial conditional invariant regularization, as it induces conditional distribution invariance over only one instantiation of W, contrary to other MMD based approaches [B,C]”, which have access to all instances of W. “*
>
>
> **3. (A) It is not clear that partial conditioning on W = 0 (apply MMD only on normal data) is enough for obtaining invariant representations in the confounded case.**
> Please refer to the global response, point 3.
>
> **Changes:** We add a sub-section to the discussion where we address to limitation of partial invariance conditioning in AD, and where summarize the insights presented in the global response.
>
> **3. (B) It is noted that “domain shifts are stronger than covariate shifts” but this is not formalized anywhere nor is it considered in the method. How important is this fact for the overall framework?**
> This was an observation mentioned in a previous work [R. Geirhos et. al. (2020) "Shortcut learning in deep neural networks."] and we added it for completeness. We agree that in our context this is not relevant and we can remove it in the next version of the paper.
>
>
> **4. The description in the background section (3.1) does not seem to match the causal graphs in Figure 2. “Conceptually, X_a represents the component of X that determines whether X is an anomaly, while X_e comprises style features. These style features, while unaffected by the anomaly status of an object, are influenced by the environment.” This description would suggest that the anomaly status (W) will affect only X_a but not X_e, but we have an arrow between W and X_e in Figure 2. Can the authors clarify this?**
>
> The reviewer is right. The sentence “while unaffected by the anomaly status of an object, are influenced by the environment.” is incorrect. The style features are also influenced by the environment, as Figure 2 shows.
>
> **Changes:** We will update this sentence as follows: *"Conceptually, X_a represents the component of X that determines whether X is an anomaly, while X_e comprises style features."*

---

> > ### Comment · Reviewer_zVDG · 2023-08-18
> > **Post rebuttal comments**
> >
> > I thank the authors for their rebuttal. I think that having a discussion about partial conditioning on W=0 will improve the paper. Showing that partial conditioning works in practice is a good contribution as is the discussion about its limitations.
> >
> > Q: In the new experiments, what is the performance of the IRM and LISA for the pretraining task? Are the poor results on the AD task due to poor features learned by IRM / LISA?
> >
> > Overall, I recommend acceptance and I will increase my score to 7.

---

> > > ### Author Response · Authors · 2023-08-19
> > > **Response to reviewer zVDG**
> > >
> > > We greatly appreciate the reviewer's feedback and the update to the score.
> > >
> > > We acknowledge the recommendation to include a discussion on partial conditioning on $W=0$, which will be added to the final manuscript following the proposed changes in the original rebuttal.
> > >
> > > **Regarding the performance of IRM and LISA for the pretraining task:**
> > > In our experiments, we set up a classification task using all classes except the one used as an anomaly. For instance, in the MNIST setting where '4' is treated as an anomaly, all other digits are used in a classification task for pretraining the encoder. The out-of-distribution accuracy obtained in MNIST during the pre-training task was 69.2% for IRM and 73.8% for LISA. In the F-MNIST setting, the accuracy was 65.3% for IRM and 68.4% for LISA. These are strong enough out-of-distribution results for us to believe that the models learned meaningful features.
> > >
> > > We suspect that while invariant features might be adequate for instilling sufficient invariance initially, these features can be compromised during the fine-tuning phase of other AD detectors, especially in the absence of any continuous incentive for maintaining invariance throughout the training process. This could justify its drop in performance from in-distribution to out-of-distribution. Also, when compared to encoders pre-trained in larger datasets such as ImageNet (or even relying on ERM in the same dataset), the quality of the features could not be as meaningful/generalizable to a different task as AD. This could justify why even in-distribution, the anomaly detectors pretrained with IRM and LISA cannot perform on par with the original baselines.
> > >
> > > We look forward to any further comments or suggestions, and once again, thank you for your constructive feedback and engaged discussion.

---

### Author Rebuttal · Authors · 2023-08-10

We thank all reviewers for their time and constructive feedback when reviewing this work.

In this global response, we will provide details on the additional experiments performed and address the general concern of partial conditioning on only W=0.

**1. Experiments on Waterbird**
To address the request of the reviewers for an additional experimental setting, we have also evaluated our method in the Waterbird dataset [A]. This is a real-world natural image dataset where the distribution shift occurs as the natural habit depicted in the background changes between an aquatic and land setting. From the two kinds of birds in the dataset (water and land birds), water birds were assigned to training data and land birds were set as an anomaly. To make this a more challenging setup, we have defined a highly unbalanced distribution of the environments, with 184 images in training data with a water background and 3498 with a land background. The evaluation of the methods was performed in images with only a water background for out-of-distribution and only land background for in distribution.

Looking at the Fig. 1 in the attached .pdf we observe that in all original unregularized methods evaluated, there is a drop in performance when comparing in-distribution (background transparent plots) and out-of-distribution (foreground opaque plots). The performance gained by adding partial conditional invariant regularization to each method ranges from 5% to 15% AUROC.

**Changes:** These experiments will be added to the main manuscript.

[A] Sagawa, Shiori, et al. "Distributionally robust neural networks for group shifts: On the importance of regularization for worst-case generalization." ICLR 2020

**2. Experiments with pre-trained invariant encoders**
We also extend our experimental evaluation by adding a comparison to [B], an environment-aware framework for AD that pretrains the encoder of the AD model using an invariance-inducing method (LISA or IRM). We evaluate this method in MNIST and F-MNIST subject to targeted covariate shifts (the exact same setup as seen in our original experiments). The method was incorporated into all baselines (SOTA AD methods) and compared to the same baselines while regularized through partial conditional invariance.

We show the results of these experiments in Fig. 2 of the attached .pdf.
Across all methods tested, and in both datasets, we observe a sharp decrease in performance when compared to a baseline non-invariant method and that it provides less robustness to covariate shifts than our proposed methodology.

[B] Smeu et al. "Env-aware anomaly detection: Ignore style changes, stay true to content!." arXiv (2022)

**Changes:** These experiments will be added to the main manuscript, and the framework described in the Related Work section.

**3. Additional details on partial conditioning (W=0)**
It is unclear if partial conditioning on W=0 is enough for all settings of distribution shifts AD. We agree that partial conditioning on W=1 would be better, however, this is not possible as partial conditioning on W=1 requires a representative sample of anomalies.

Note, however, that even partial conditioning on both W = 0 and W = 1 would not be enough to attain invariant representations, from a theoretical point of view. The state of the art has only been able to provide necessary conditions for learning invariant representations. This is reflected by Corollary 1, where we only offer necessary conditions for invariant representations. This corollary derives from the recent works from (Veitch et al). Sufficient conditions for learning invariant representations are still unknown and remain a challenge for future work.

Still, we want to provide intuition for cases where partial conditioning is enough. We believe that the main condition to understand the limitation of the invariance of (W=0) is the behaviour of the style feature across the two instantiations of W. In particular, how disentangled these two variables are in terms of their information content. For example, in MNIST, in a setting where the colour of the background changes, both normal images and abnormal images will have the information concerning their digit remain the same. Therefore, in this setting, if these two variables are captured well enough by the un-regularised AD method, attaining invariance to the background colour with W=0 could also lead to robust behaviour for W=1. To some extent, we also hypothesize that the limitation of the partial conditional invariance is dependent on how the final representation achieved by the encoder is able to disentangle the factors from $X_e$ and $X_a$, and could see improvements as general representing learning advances.

Furthermore, we believe there is an inherent benefit to being agnostic to the outlier process (W=1) when inducing an invariance condition in AD, as any assumptions concerning the characterization of an anomaly are not falsifiable during the training process.

Finally, one of our contributions is to demonstrate empirically that learning representations with just partial conditioning on W=0 is still an effective way of attaining invariant representations. Camelyon17 is a dataset where the mechanisms connecting the environment variable and the abnormal class are less clear, in particular, how the physiological characteristics of the population in the out-of-distribution environment affect the tumorous case. However, in this setting, we are still able to considerably improve the robustness of all evaluated models by only relying on partial conditioning on W=0.

**Changes:** We will add a sub-section to the discussion with the limitations of partial conditioning (W=0).

**Final remarks**
We believe to have addressed all questions and weaknesses raised during the review process. We would greatly appreciate it if the reviewers responded to the proposed change, and would kindly ask you to adjust your review score while taking the rebuttal into account.

---

### Decision · Program_Chairs · 2023-09-21

**Decision:**

Accept (poster)

**Comment:**

The paper addresses the problem of anomaly detection under distribution shift. Three of the reviewers (zVDG, yiE8, ZicU) have chosen to accept the paper, with various degrees of enthusiasm. The reviewers appreciated the way in which the problem was formalized and motivated, the use of the MMD regularization, with one of the reviewers seeing the approach as creative. The method was also noted by a different reviewer as "pioneer in studying OOD detection from the causal perspective", an assessment which I have come to agree with. Two of the reviewers (ePdN, AUeF) have chosen to borderline reject, with the discussion not changing their opinion. On the other hand, the reviewers questioned whether other existing AD methods could be better adapted for the same setting. A concern that was brought up was the similarity to Red PANDA.The authors made a convincing case, in the discussion, that the theoretical justification for the partial conditional invariance is nontrivial, and that the method goes beyond simply handling spurious correlations and also outlined the differences between their approach and Red PANDA/DCoDR, which I found to be compelling. This is a new method, addressing a very specific problem that was previously not fully considered, let alone solved.

Finally, as the reviewers have not reached consensus following the discussion period, I read the paper myself. I found that it clearly illustrates the problem, makes a compelling case why invariance should be a factor in training anomaly detection models, presents a pertinent method to solve the problem and demonstrates its use on a number of datasets. As such, it should be of interest and of potential use to a segment of the NeurIPS community.